# Sequential Order-Robust Mamba for Time Series Forecasting

## Abstract

Mamba has recently emerged as a promising alternative to Transformers, offering near-linear complexity in processing sequential data. However, while channels in time series (TS) data have no specific order in general, recent studies have adopted Mamba to capture channel dependencies (CD) in TS, introducing a *sequential order bias*. To address this issue, we propose SOR-Mamba, a TS forecasting method that 1) incorporates a regularization strategy to minimize the discrepancy between two embedding vectors generated from data with reversed channel orders, thereby enhancing robustness to channel order, and 2) eliminates the 1D-convolution originally designed to capture local information in sequential data. Furthermore, we introduce channel correlation modeling (CCM), a pretraining task aimed at preserving correlations between channels from the data space to the latent space in order to enhance the ability to capture CD. Extensive experiments demonstrate the efficacy of the proposed method across standard and transfer learning scenarios.

## 1 Introduction

Time series (TS) forecasting is prevalent in various fields, including weather (Angryk et al., 2020), traffic (Cirstea et al., 2022), and energy (Dudek et al., 2021). While Transformers (Vaswani et al., 2017) have been widely employed for this task due to their ability to capture long-term dependencies in sequences (Wen et al., 2022), their quadratic computational complexity causes substantial computational overhead, limiting their practicality in real-world applications. Several attempts have been made to reduce the complexity of Transformers (Zhang & Yan, 2023; Zhou et al., 2022); however, they often result in performance degradations (Wang et al., 2024).

To tackle the computational challenges of Transformers, alternatives such as state-space models (SSMs) (Gu et al., 2022) have been considered, employing convolutional operations to process sequences with linear complexity. Recently, Mamba (Gu & Dao, 2023) enhanced SSMs by incorporating a selective mechanism to prioritize important information efficiently. Due to its strong balance between performance and computational efficiency (Wang et al., 2024), Mamba has been widely adopted across various domains (Zhu et al., 2024; Schiff et al., 2024). In the TS domain, Mamba is utilized to capture temporal dependencies (TD) by processing input TS along the *temporal dimension* (Ahamed & Cheng, 2024), channel dependencies (CD) along the *channel dimension* (Wang et al., 2024), or both (Cai et al., 2024). In this paper, we focus on Mamba capturing CD, in line with the recent work (Liu et al., 2024a) that advocates for the use of complex attention mechanisms for CD while employing simple multi-layer perceptrons (MLPs) for TD.

However, applying Mamba to capture CD is challenging as channels lack an inherent sequential order, whereas Mamba is originally designed for sequential inputs (i.e., Mamba contains a *sequential order bias*), as shown in Figure 1. To address this issue, previous works have employed the bidirectional Mamba to capture CD (Wang et al., 2024; Liang et al., 2024), where two unidirectional Mambas with different parameters capture CD from a certain channel order and its reversed order. However, these methods are inefficient due to the need for two models. Another approach

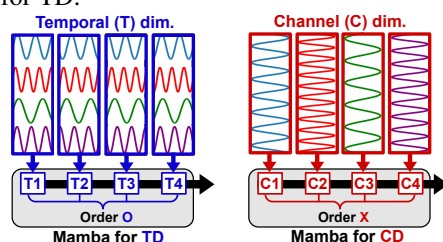

Figure 1: Capturing CD with Mamba, which has a sequential order bias, is challenging as channels lack an inherent sequential order.

involves permuting a channel order during training (Cai et al., 2024) to enhance robustness to the order, while requiring an additional procedure to determine the optimal order for inference.

Furthermore, Table 1 shows the performance of the TS forecasting task using the bidirectional Mamba and two unidirectional Mambas with reversed channel orders, suggesting that the bidirectional Mamba (Wang et al., 2024) may not be effective in handling the sequential order bias. The table indicates that 1) the bidirectional Mamba does not always achieve the best performance, and 2) the performance of the unidirectional Mamba varies depending on the channel order.

| ECL dataset (Metric: MSE) | Horizon | | | |
|---|---|---|---|---|
| | 96 | 192 | 336 | 720 |
| Bidirectional | **0.139** | 0.165 | **0.177** | 0.214 |
| ① Uni ($1 \rightarrow C$) | 0.143 | **0.162** | 0.179 | 0.234 |
| ② Uni ($C \rightarrow 1$) | 0.141 | 0.168 | 0.179 | **0.210** |
| ((① - ②) / ①) | +1.6% | -3.8% | -0.2% | +10.3% |

Table 1: Bidirectional Mamba may not achieve the best performance, and the performance of the unidirectional Mamba varies by the channel order.

To this end, we introduce **S**equential **O**rder-**R**obust **Mamba** for TS forecasting (*SOR-Mamba*), a TS forecasting method that handles the sequential order bias by 1) incorporating a regularization strategy to minimize the distance between two embedding vectors generated from data with reversed channel orders to enhance robustness to the order, and 2) removing the 1D-convolution (1D-conv) originally designed to capture local information in sequential inputs. Additionally, we propose **C**hannel **C**orrelation **M**odeling (*CCM*), a pretraining task aimed at improving the model's ability to capture CD by preserving the correlation between channels from the data space to the latent space. The main contributions of this work are summarized as follows:

- We propose SOR-Mamba, a TS forecasting method that handles the sequential order bias by 1) regularizing the unidirectional Mamba to minimize the distance between two embedding vectors generated from data with reversed channel orders for robustness to channel order and 2) removing the 1D-conv from the original Mamba block, as channels lack an inherent sequential order.

- We introduce CCM, a novel pretraining task that preserves the correlation between channels from the data space to the latent space, thereby enhancing the model's ability to capture CD.

- We conduct extensive experiments with 13 datasets in both standard and transfer learning settings, demonstrating that our method achieves state-of-the-art (SOTA) performance with greater efficiency compared to previous SOTA methods by utilizing the unidirectional Mamba.

## 2 RELATED WORKS

**TS forecasting with Transformer.** Transformers (Vaswani et al., 2017) are commonly employed for long-term TS forecasting (LTSF) tasks due to their ability to handle long-range dependencies through attention mechanisms. However, their quadratic complexity has led to the development of various methods aimed at improving efficiency, such as modifying the Transformer architecture (Zhang & Yan, 2023; Zhou et al., 2022), patchifying the TS (Nie et al., 2023) or using MLP-based models (Chen et al., 2023; Zeng et al., 2023). While MLP-based models offer simpler structures and reduced complexity compared to Transformers, they tend to be less effective at capturing global dependencies (Wang et al., 2024). Recently, iTransformer (Liu et al., 2024a) inverts the conventional Transformer framework in the TS domain by treating each channel as a token rather than each patch, shifting the focus from capturing TD to CD. This framework has led to significant performance gains and has become widely adopted as the backbone for TS models (Liu et al., 2024b; Dong et al., 2024).

**State-space models.** To overcome the limitations of Transformer-based models, state-space models have been integrated with deep learning to tackle the challenge of long-range dependencies (Rangapuram et al., 2018; Zhang et al., 2023; Zhou et al., 2023). However, these methods are unable to adapt their internal parameters to varying inputs, which limits their performance. Recently, Mamba (Gu & Dao, 2023) introduces a selective scan mechanism that efficiently filters specific inputs and captures long-range context by incorporating time-varying parameters into the SSM. Due to its linear-time efficiency for modeling long sequences, it has been widely adopted in various domains, including computer vision (Ma et al., 2024a; Huang et al., 2024; Zhu et al., 2024) and natural language processing (Pióro et al., 2024; Anthony et al., 2024; He et al., 2024).

**TS forecasting with Mamba.** Due to its balance between performance and computational efficiency, Mamba has also been applied in the TS domain. TimeMachine (Ahamed & Cheng, 2024) utilizes multi-scale quadruple-Mamba to capture either TD alone or both TD and CD, with its architecture relying on the statistics of the dataset. CMamba (Zeng et al., 2024) captures TD with patch-wise Mamba and CD with an MLP. FMamba (Ma et al., 2024b) integrates fast-attention with Mamba to capture CD, and SST (Xu et al., 2024) captures global and local patterns in TS with Mamba and Transformer, respectively. S-Mamba (Wang et al., 2024), Bi-Mamba+ (Liang et al., 2024), and

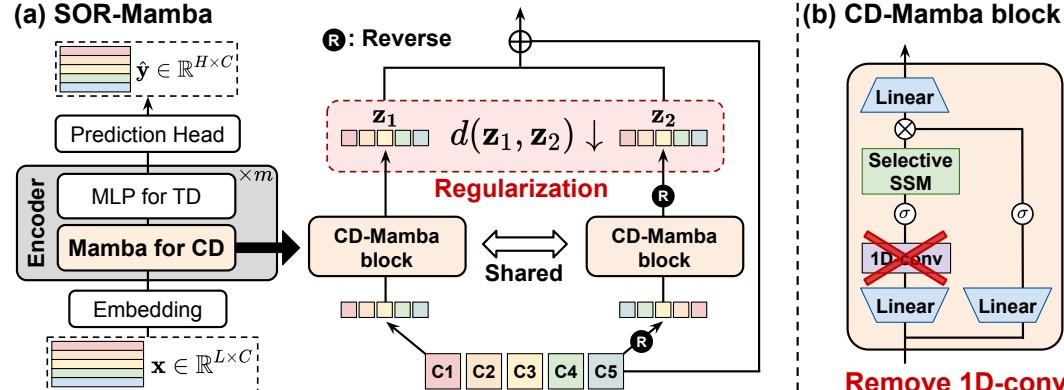

Figure 2: **Overall framework of SOR-Mamba.** (a) shows the architecture of SOR-Mamba, where the CD-Mamba block is regularized to minimize the distance between two vectors derived from reversed channel orders. (b) shows the CD-Mamba block, where the 1D-conv from the Mamba block is removed, as channels do not have a sequential order, which is further explained in Appendix D.

SAMBA (Weng et al., 2024), designed to capture CD in TS, use bidirectional scanning with the bidirectional Mamba to address the sequential order bias, although they are limited by the need for two models. MambaTS (Cai et al., 2024) introduces variable permutation training, which shuffles the channel order during the training stage to handle the sequential order bias. However, it is limited by the need for an additional procedure to determine the optimal scan order for the inference stage.

## 3 PRELIMINARIES

**Problem definition.** This paper addresses the multivariate TS forecasting task, where the model uses a lookback window $\mathbf{x} = (\mathbf{x}_1, \mathbf{x}_2, \cdots, \mathbf{x}_L)$ to predict future values $\mathbf{y} = (\mathbf{x}_{L+1}, \cdots, \mathbf{x}_{L+H})$ with $\mathbf{x}_i \in \mathbb{R}^C$ representing the values at each time step. Here, $L$, $H$, and $C$ denote the size of the lookback window, the forecast horizon, and the number of channels, respectively.

**State-space models.** SSM transforms the continuous input signals $x(t)$ into corresponding outputs $y(t)$ via a state representation $h(t)$. This state space represents how the state evolves over time, which can be expressed using ordinary differential equations as follows:

$$
\begin{aligned}
h'(t) &= \boldsymbol{A}h(t) + \boldsymbol{B}x(t), \\
y(t) &= \boldsymbol{C}h(t) + \boldsymbol{D}x(t),
\end{aligned}
\tag{1}
$$

where $h'(t) = \frac{dh(t)}{dt}$, and $\boldsymbol{A}, \boldsymbol{B}, \boldsymbol{C}$, and $\boldsymbol{D}$ are learnable parameters of the SSMs.

Due to the continuous nature of SSMs, discretization is commonly used to approximate continuous-time representations into discrete-time representations by sampling input signals at fixed intervals. This results in the discrete-time SSMs being represented as:

$$
\begin{aligned}
h_k &= \overline{\boldsymbol{A}}h_{k-1} + \overline{\boldsymbol{B}}x_k, \\
y_k &= \boldsymbol{C}h_k + \boldsymbol{D}x_k,
\end{aligned}
\tag{2}
$$

where $h_k$ and $x_k$ are the state vector and input vector at time $k$, respectively, and $\overline{\boldsymbol{A}} = \exp(\Delta \boldsymbol{A})$ and $\overline{\boldsymbol{B}} = (\Delta \boldsymbol{A})^{-1}(\exp(\Delta \boldsymbol{A}) - \boldsymbol{I}) \cdot \Delta \boldsymbol{B}$ are the discrete-time matrices obtained from the $A$ and $B$.

Recently, Mamba introduces selective SSMs that enables the model to capture contextual information in long sequences using time-varying parameters (Gu & Dao, 2023). Its near-linear complexity makes it an efficient alternative to the quadratic complexity of the attention mechanism in Transformers.

## 4 METHODOLOGY

In this paper, we introduce SOR-Mamba, a TS forecasting method designed to address the sequential order bias by 1) regularizing Mamba to minimize the distance between two embedding vectors generated from data with reversed channel orders and 2) removing the 1D-conv from the original Mamba block. The overall framework of SOR-Mamba is illustrated in Figure 2, which consists of four components: the embedding layer for tokenization, Mamba for capturing CD, MLP for capturing TD, and the prediction head for predicting the future output.

Furthermore, we introduce a novel pretraining task, CCM, where the model is pretrained to preserve the correlation between channels from the data space to the latent space, aligning with the recent TS models that focus on capturing CD over TD. The overall framework of CCM is illustrated in Figure 3.

## 4.1 ARCHITECTURE OF SOR-MAMBA

**1) Embedding layer.** To tokenize the TS in a channel-wise manner, we use an embedding layer that treats each channel as a token, following the approach in iTransformer (Liu et al., 2024a). Specifically, we transform $\mathbf{x} \in \mathbb{R}^{L \times C}$ into $\mathbf{z} \in \mathbb{R}^{C \times D}$ using a single linear layer.

**2) Mamba for CD.** The original Mamba block combines the H3 block (Fu et al., 2023) with a gated MLP, where the H3 block incorporates a 1D-conv before the SSM layer to capture local information from adjacent steps. However, since channels in TS do not possess any inherent sequential order, we find this convolution unnecessary for capturing CD. Accordingly, we remove the convolution from the original Mamba block, resulting in the proposed *CD-Mamba block*, as illustrated in Figure 2(b). Note that this differs from the previous work (Cai et al., 2024) which replaces the 1D-conv with a dropout in the Mamba block, as it is designed to capture TD. Using the CD-Mamba block, we obtain $\mathbf{z}_1$ and $\mathbf{z}_2$, which are two embedding vectors with reversed channel orders that are employed for regularization to address the sequential order bias. These vectors are then added element-wise and combined with a residual connection from $\mathbf{z}$. Further analysis regarding the removal of the 1D-conv can be found in Table 7.

**3) MLP for TD.** To capture TD in TS, we apply an MLP to the output tokens of the CD-Mamba block. To enhance training stability, we apply layer normalization (LN) to standardize the tokens both before and after the MLP.

**4) Prediction head.** To predict the future output, we employ a linear prediction head to the output tokens of MLP, resulting in $\hat{\mathbf{y}} \in \mathbb{R}^{H \times C}$. The procedure of SOR-Mamba is described in Algorithm 2, where $\mathbf{Z}^\star$ represents $\mathbf{Z}$ with its channel order reversed.

---

**Algorithm 1** The procedure of SOR-Mamba

**Input**: $\mathbf{X} = [\mathbf{X}_1, \ldots, \mathbf{X}_L] : (B, L, C)$
**Output**: $\hat{\mathbf{Y}} = [\hat{\mathbf{X}}_{L+1}, \ldots, \hat{\mathbf{X}}_{L+H}] : (B, H, C)$
1: $\mathbf{Z} : (B, C, D) \leftarrow \text{Linear}(\mathbf{X}^\top)$
2: **for** $m$ in layers **do**
3:      $\mathbf{Z}_1 : (B, C, D) \leftarrow \text{CD-Mamba}(\mathbf{Z})$
4:      $\mathbf{Z}_2 : (B, C, D) \leftarrow \text{CD-Mamba}(\mathbf{Z}^\star)^\star$,
                 where $\mathbf{Z}^\star = \mathbf{Z}[:, :: -1, :]$
5:      $\mathbf{Z} : (B, C, D) \leftarrow (\mathbf{Z}_1 + \mathbf{Z}_2) + \mathbf{Z}$
6:      $\mathbf{Z} : (B, C, D) \leftarrow \text{LN}(\text{MLP}(\text{LN}(\mathbf{Z})))$
7: **end for**
8: $\hat{\mathbf{Y}} : (B, H, C) \leftarrow \text{Linear}(\mathbf{Z})^\top$

---

## 4.2 REGULARIZATION WITH CD-MAMBA BLOCK

To address the sequential order bias, SOR-Mamba regularizes the CD-Mamba block to minimize the distance between two embedding vectors generated from data with reversed channel orders. The regularization term is defined as follows:

$$L_{\text{reg}}(\mathbf{z}) = d(\mathbf{z}_1, \mathbf{z}_2), \tag{3}$$

where $d$ is a distance metric, and $\mathbf{z}_1$ and $\mathbf{z}_2$ are the embedding vectors obtained from the CD-Mamba block using $\mathbf{z}$ with its channel order reversed, as described in Algorithm 2. For $d$, we use the mean squared error (MSE) in the experiments, where the robustness to the choice of $d$ can be found in Appendix K. The proposed regularization term is then added to the forecasting loss ($L_{\text{fcst}}$) with a contribution of $\lambda$, resulting in:

$$L(\mathbf{x}, \mathbf{y}) = L_{\text{fcst}}(\mathbf{x}, \mathbf{y}) + \lambda \cdot \sum_{i=1}^{m} L_{\text{reg}}(\mathbf{z}^{(i)}), \tag{4}$$

where $\mathbf{z}^{(i)}$ is $\mathbf{z}$ at the $i$-th layer, and $m$ is the number of encoder layers. By incorporating the regularization strategy into the unidirectional Mamba, we achieve better performance and efficiency compared to S-Mamba (Wang et al., 2024), which employs the bidirectional Mamba, as shown in Table 5. Additionally, we find that regularization also benefits the bidirectional Mamba, which handles the sequential order bias through bidirectional scanning, as shown in Table 6. Further analysis regarding the robustness to $\lambda$ is discussed in Appendix I.

## 4.3 CHANNEL CORRELATION MODELING

Previous pretraining tasks for TS have primarily focused on TD, such as masked modeling (Zerveas et al., 2021) and reconstruction (Lee et al., 2024), to pretrain an encoder. However, we argue for the necessity of a new task that emphasizes CD over TD to align with recent TS models that focus on capturing CD with complex model architectures (Liu et al., 2024a; Wang et al., 2024). To this end, we propose CCM, which aims to preserve the (Pearson) correlation between channels from the data space to the latent space, as correlation is a simple yet effective way to measure channel relationships and has been utilized in prior studies to analyze CD (Yang et al., 2024; Zhao & Shen, 2024).

| Models | (1) Mamba | | | | | | (2) Transformer | | | | | | (3) Linear/MLP | | | | | |
|---|---|---|---|---|---|---|---|---|---|---|---|---|---|---|---|---|---|---|
| | SOR-Mamba | | | | S-Mamba | | iTransformer | | PatchTST | | Crossformer | | TimesNet | | DLinear | | RLinear | |
| | FT | | SL | | | | | | | | | | | | | | | |
| Metric | MSE | MAE | MSE | MAE | MSE | MAE | MSE | MAE | MSE | MAE | MSE | MAE | MSE | MAE | MSE | MAE | MSE | MAE |
| ETTh1 | **.433** | .436 | .442 | .438 | .457 | .452 | .454 | .449 | .469 | .454 | .529 | .522 | .458 | .450 | .456 | .452 | .446 | **.434** |
| ETTh2 | .376 | .405 | .382 | .407 | .383 | .408 | .384 | .407 | .387 | .407 | .942 | .684 | .414 | .427 | .559 | .515 | **.374** | **.398** |
| ETTm1 | .391 | **.400** | .396 | .401 | .398 | .407 | .408 | .412 | **.387** | **.400** | .513 | .496 | .400 | .406 | .403 | .407 | .414 | .407 |
| ETTm2 | **.281** | .327 | .284 | .329 | .290 | .333 | .293 | .337 | **.281** | **.326** | .757 | .610 | .291 | .333 | .350 | .401 | .286 | .327 |
| PEMS03 | **.121** | **.227** | .137 | .242 | .133 | .240 | .142 | .248 | .180 | .291 | .169 | .281 | .147 | .248 | .278 | .375 | .495 | .472 |
| PEMS04 | .099 | **.203** | .107 | .212 | **.096** | .205 | .121 | .232 | .195 | .307 | .209 | .314 | .129 | .241 | .295 | .388 | .526 | .491 |
| PEMS07 | **.088** | **.186** | .091 | .191 | .090 | .191 | .102 | .205 | .211 | .303 | .235 | .315 | .124 | .225 | .329 | .395 | .504 | .478 |
| PEMS08 | **.142** | **.232** | .162 | .247 | .157 | .242 | .254 | .306 | .280 | .321 | .268 | .307 | .193 | .271 | .379 | .416 | .529 | .487 |
| Exchange | .358 | **.402** | .363 | .405 | .364 | .407 | .368 | .409 | .367 | .404 | .940 | .707 | .416 | .443 | **.354** | .414 | .378 | .417 |
| Weather | .256 | **.277** | .257 | .278 | **.252** | **.277** | .260 | .281 | .259 | .281 | .259 | .315 | .259 | .287 | .265 | .317 | .272 | .291 |
| Solar | .230 | .259 | .242 | .274 | .244 | .275 | .234 | .261 | .270 | .307 | .641 | .639 | .301 | .319 | .330 | .401 | .369 | .356 |
| ECL | **.168** | .264 | .169 | **.262** | .174 | .269 | .179 | .270 | .205 | .290 | .244 | .334 | .192 | .295 | .212 | .300 | .219 | .298 |
| Traffic | **.402** | **.273** | .412 | .276 | .417 | .277 | .428 | .282 | .481 | .304 | .550 | .304 | .620 | .336 | .625 | .383 | .626 | .378 |
| Average | **.257** | **.299** | .265 | .305 | .266 | .307 | .278 | .315 | .306 | .338 | .481 | .448 | .303 | .329 | .372 | .397 | .418 | .403 |
| 1st Count | 33 | 31 | 7 | 10 | 10 | 7 | 1 | 3 | 8 | 7 | 3 | 0 | 0 | 0 | 2 | 0 | 3 | 9 |
| 2nd Count | 15 | 19 | 18 | 19 | 13 | 13 | 9 | 6 | 1 | 6 | 0 | 0 | 0 | 1 | 2 | 0 | 2 | 2 |

Table 2: **Results of multivariate TS forecasting.** We compare our method with the SOTA methods under both SL and SSL settings. The best results are in **bold** and the second best are underlined.

For CCM, we calculate the correlation matrices between the input token on the data space and the output token after the additional linear projection layer on the latent space, as shown in Figure 3. The loss function for CCM, defined as the distance between these two matrices, can be expressed as:

$$L_{\text{CCM}}(\mathbf{x}) = d\left(\mathbf{R_x}, \mathbf{R_z}\right), \tag{5}$$

where $\mathbf{R_x}$ and $\mathbf{R_z}$ are the correlation matrices in the data space and the latent space, respectively. We find that CCM is more effective than masked modeling and reconstruction across diverse datasets with varying numbers of channels, as demonstrated in Table 9. Additionally, robustness to the choice of $d$ and the pseudocode of CCM are discussed in Appendix K and Appendix J, respectively.

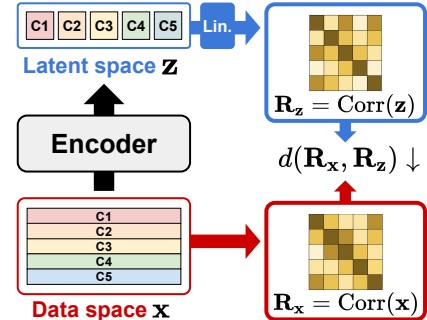

Figure 3: Channel correlation modeling.

## 5 EXPERIMENTS

### 5.1 EXPERIMENTAL SETTINGS

**Tasks and evaluation metrics.** We demonstrate the effectiveness of SOR-Mamba on TS forecasting tasks with 13 datasets under standard and transfer learning settings. For evaluation, we follow the standard self-supervised learning (SSL) framework, which involves pretraining and fine-tuning (FT) or linear probing (LP) on the same dataset. Additionally, we consider in-domain and cross-domain transfer learning settings, with the domains defined in the previous work (Dong et al., 2023). For evaluation metrics, we employ mean squared error (MSE) and mean absolute error (MAE).

**Datasets.** For the forecasting tasks, we use 13 datasets: four ETT datasets (ETTh1, ETTh2, ETTm1, ETTm2) (Zhou et al., 2021), four PEMS datasets (PEMS03, PEMS04, PEMS07, PEMS08) (Chen et al., 2001), Exchange, Weather, Traffic, Electricity (ECL) (Wu et al., 2021), and Solar-Energy (Solar) (Lai et al., 2018). Details of the dataset statistics are provided in Appendix A.

**Baseline methods.** We follow the baseline methods and results from S-Mamba (Wang et al., 2024). For the baseline methods, we consider Transformer-based models, including iTransformer (Liu et al., 2024a), PatchTST (Nie et al., 2023), and Crossformer (Zhang & Yan, 2023), as well as linear/MLP models, including TimesNet (Wu et al., 2023), DLinear (Zeng et al., 2023), and RLinear (Li et al., 2023). Additionally, we include S-Mamba (Wang et al., 2024), which is a Mamba-based TS forecasting model. Details of the baseline methods are provided in Appendix B.

**Experimental setups.** We follow the experimental setups from iTransformer and S-Mamba. Note that we do not tune any hyperparameters except for $\lambda$, which is related to the proposed regularization, while adhering to the values used in S-Mamba for all other hyperparameters concerning the model architecture and optimization. For dataset splitting, we adhere to the standard protocol of dividing all datasets into training, validation, and test sets in chronological order. Details of the experimental setups, including the size of the input window and the forecast horizon, are provided in Appendix A.

| Dataset | SL | SSL (CCM) | |
|---|---|---|---|
| | | LP | FT |
| ETTh1 | .442 | .452 | **.433** |
| ETTh2 | .382 | **.376** | **.376** |
| ETTm1 | .396 | .399 | **.391** |
| ETTm2 | .284 | .283 | **.281** |
| Exchange | .363 | **.349** | .358 |
| Solar | .242 | **.230** | **.230** |
| ECL | .169 | .169 | **.168** |

Table 3: SL vs. SSL.

| | Source | Target | SOR-Mamba | | | S-Mamba | | |
|---|---|---|---|---|---|---|---|---|
| | | | SL | LP | FT | SL | LP | FT |
| In-domain | ETTh2 | ETTh1 | .442 | .452 | **.433** | .457 | .450 | .464 |
| | ETTm2 | ETTm1 | .396 | .401 | **.390** | .398 | .398 | .400 |
| | Average | | .419 | .427 | **.411** | .428 | .425 | .432 |
| Cross-domain | ETTm2 | ETTh1 | .442 | .448 | **.433** | .457 | .450 | .455 |
| | ETTh2 | ETTm1 | .396 | .399 | **.391** | .398 | .401 | .402 |
| | ETTm1 | ETTh1 | .442 | .449 | **.434** | .457 | .450 | .468 |
| | ETTh1 | ETTm1 | .396 | .404 | **.391** | .398 | .403 | .399 |
| | Weather | ETTh1 | **.442** | .545 | .542 | .457 | .546 | .552 |
| | Weather | ETTm1 | **.396** | .457 | .458 | .398 | .460 | .501 |
| | Average | | **.419** | .450 | .441 | .428 | .452 | .463 |

Table 4: Results of transfer learning.

| Improvements | (1) Performance | | | | | | (2) Efficiency | |
|---|---|---|---|---|---|---|---|---|
| | Average MSE across four horizons | | | | Average | | # Params. | Impr. |
| | ETTh1 | ETTh2 | ETTm1 | ETTm2 | MSE | Impr. | | |
| S-Mamba | .457 | .383 | .398 | .290 | .382 | - | 9.29M | - |
| + Regularization | .452 | .382 | .394 | .286 | .378 | 1.0% | 9.29M | - |
| + Bi → Unidirectional | .449 | .382 | .396 | .285 | .378 | 0.1% | 5.81M | 37.5% |
| + Remove 1D-conv | .442 | .382 | .396 | .284 | .376 | 0.5% | **5.80M** | 0.1% |
| + CCM | **.433** | **.376** | **.391** | **.281** | **.370** | 1.5% | **5.80M** | - |

Table 5: Ablation study of  Regularization ,  Model architecture  and  Pretraining task .

## 5.2 Time Series Forecasting

Table 2 presents the comprehensive results for the multivariate TS forecasting task, showing the average MSE/MAE across four horizons over five runs. The results demonstrate that our proposed SOR-Mamba outperforms the SOTA Transformer-based models and S-Mamba, which uses the bidirectional Mamba, whereas our approach utilizes the unidirectional Mamba, providing greater efficiency as discussed in Table 13. Furthermore, self-supervised pretraining (SSL) with CCM yields additional performance gains compared to the supervised setting (SL), with comparisons to SL and SSL (LP and FT) shown in Table 3. Full results of Table 2 are provided in Appendix E.

## 5.3 Transfer Learning

To assess the transferability of our method, we conduct transfer learning experiments in both in-domain and cross-domain transfer settings following SimMTM (Dong et al., 2023), where source and target datasets share the same frequency in the in-domain setting, while they do not in the cross-domain setting. Table 4 presents the average MSE across four horizons, demonstrating that SOR-Mamba consistently outperforms S-Mamba, achieving nearly a 5% performance gain in FT.

## 5.4 Ablation Study

To demonstrate the effectiveness of our method, we conduct an ablation study using four ETT datasets to evaluate the impact of the following components: 1) adding the regularization term, 2) using the unidirectional Mamba instead of the bidirectional Mamba, 3) removing the 1D-conv, and 4) pretraining with CCM. Table 5 presents the results, indicating that using all proposed components results in the best performance and that our method outperforms S-Mamba with 37.6% fewer model parameters. The full results of the ablation study are provided in Appendix F.

## 6 Analysis

**Sequential order bias.** The degree of a sequential order bias may vary depending on the characteristics of the datasets. We consider two factors affecting this degree: 1) the *correlation between channels* and 2) the *number of channels* in the dataset. To evaluate the relationships between these factors and the degree of bias, we quantify the degree of a sequential order bias for each dataset by measuring the difference in performance (average MSE across four horizons) when the channel order is reversed, using SOR-Mamba without regularization.

| Mamba | | ETT | | | | PEMS | | | | Exchange | Weather | Solar | ECL | Traffic |
|---|---|---|---|---|---|---|---|---|---|---|---|---|---|---|
| # | Reg. | h1 | h2 | m1 | m2 | 03 | 04 | 07 | 08 | | | | | |
| Bi | ✗ | .457 | .383 | .398 | .290 | .133 | **.096** | **.090** | .157 | .364 | **.252** | **.244** | .174 | .417 |
| | ✓ | **.452** | **.382** | **.394** | **.286** | **.131** | **.096** | .092 | **.155** | **.361** | **.252** | .245 | **.170** | **.411** |
| Uni | ✗ | .455 | .383 | .403 | .289 | .140 | .102 | .094 | .161 | .364 | **.255** | **.244** | .175 | **.416** |
| | ✓ | **.449** | **.382** | **.396** | **.285** | **.135** | **.101** | **.091** | **.158** | **.361** | **.255** | **.244** | **.171** | **.416** |

Table 6: **Effect of regularization.** Regularization enhances both the unidirectional and the bidirectional Mamba. Note that we do not remove the 1D-conv to isolate the effect of regularization.

| Mamba | | ETT | | | | PEMS | | | | Exchange | Weather | Solar | ECL | Traffic |
|---|---|---|---|---|---|---|---|---|---|---|---|---|---|---|
| # | 1D-conv | h1 | h2 | m1 | m2 | 03 | 04 | 07 | 08 | | | | | |
| Bi | ✓ | .457 | **.383** | .398 | .290 | **.133** | **.096** | .090 | .157 | **.364** | **.252** | .244 | .174 | .417 |
| | ✗ | **.441** | **.383** | **.396** | **.285** | .137 | .102 | **.089** | **.148** | **.364** | .255 | **.242** | **.167** | **.414** |
| Uni | ✓ | .449 | **.382** | **.396** | .285 | **.135** | **.101** | **.091** | **.158** | **.361** | **.255** | .244 | .171 | .416 |
| | ✗ | **.442** | **.382** | **.396** | **.284** | .137 | .107 | **.091** | .162 | .363 | .257 | **.242** | **.169** | **.412** |

Table 7: **Effect of 1D-conv.** Removing the 1D-conv, which captures the local information within adjacent channels, improves the performance on TS datasets that lack a sequential order in channels.

Figure 4 shows the results with two plots, where the x-axes represent the number of channels and correlation between the channels (i.e., average of the off-diagonal elements in the correlation matrix[1] between the channels), and the y-axes represent the degree of a sequential order bias, with all axes shown on a log scale. The results show that the bias increases 1) as the channels become more correlated and 2) as the number of channels increases. For example, four ETT datasets containing seven channels with low correlation show low bias, whereas four PEMS datasets containing over 100 channels with high correlation exhibit high bias.

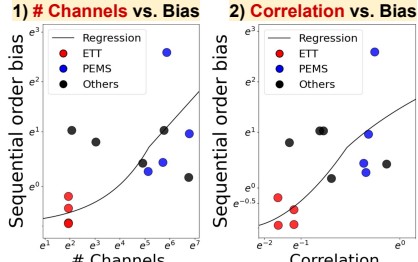

Figure 4: Varying bias across datasets.

**Effect of regularization.** To validate the effect of the regularization strategy, we apply it to both the unidirectional and the bidirectional Mamba without removing the 1D-conv to isolate the effect of regularization. The results are shown in Table 6, which presents the TS forecasting results of the average MSE across four horizons. These results indicate that it not only improves the performance of the unidirectional Mamba but also benefits the bidirectional Mamba, which handles the sequential order bias through bidirectional scanning, making regularization complementary to this approach.

**Effect of 1D-conv.** To demonstrate the unnecessity of the 1D-conv in Mamba for capturing CD, we remove it from both the unidirectional and the bidirectional Mamba, with the results of the average MSE across four horizons shown in Table 7. The results indicate that removing the 1D-conv, which captures the local information within nearby channels, improves the performance on general TS datasets where channels lack a sequential order. However, its removal may negatively impact datasets with ordered channels such as PEMS datasets (Liu et al., 2022), which consist of traffic sensor data. Figure 5 illustrates the relative gain from removing the 1D-conv in SOR-Mamba, showing that three out of four PEMS datasets achieve better results with the 1D-conv than without it.

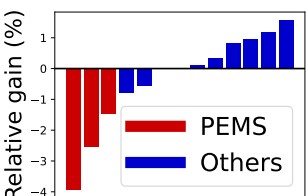

Figure 5: Effect of 1D-conv.

**Correlation for CCM.** To assess the impact of using different correlations for CCM, we consider two candidates: *local correlation*, which refers to the correlation between the channels of the input TS, and *global correlation*, which refers to the correlation between the channels of the entire TS dataset. Table 8 shows that using the local correlation yields better performance compared to the global correlation, although both approaches still outperform the supervised setting (SL).

**Effect of CCM.** To demonstrate the effect of CCM, we compare it with two other widely used pretraining tasks: masked modeling (MM)(Zerveas et al., 2021) with a masking ratio of 50%, and reconstruction (Rec.)(Lee et al., 2024), along with the supervised setting. Table 9 presents the results using two backbones, S-Mamba and SOR-Mamba, showing that CCM consistently outperforms the other tasks across both backbones.

---

[1]We use its absolute value, as high correlation does not always indicate a strong relationship, with strong negative relationships near $-1$. Additionally, we use only the off-diagonal elements to exclude autocorrelation.

| Dataset | SL | SSL (CCM) | |
|---|---|---|---|
| | | Global | Local |
| ETTh1 | .442 | .445 | **.433** |
| ETTh2 | .382 | .380 | **.376** |
| ETTm1 | .396 | .393 | **.391** |
| ETTm2 | .284 | .283 | **.281** |
| PEMS03 | .137 | .125 | **.121** |
| PEMS04 | .107 | .101 | **.099** |
| PEMS07 | .091 | **.088** | .088 |
| PEMS08 | .162 | .146 | **.142** |
| Exchange | .363 | .361 | **.358** |
| Weather | .257 | .258 | **.256** |
| Solar | .242 | **.228** | .230 |
| ECL | .169 | .170 | **.168** |
| Traffic | .412 | .410 | **.402** |
| Average | .265 | .260 | **.257** |

Table 8: Global vs. Local corr.

| Dataset | SOR-Mamba | | | | S-Mamba | | | |
|---|---|---|---|---|---|---|---|---|
| | SL | SSL | | | SL | SSL | | |
| | | Rec. | MM | CCM | | Rec. | MM | CCM |
| ETTh1 | .442 | .434 | .435 | **.433** | .457 | **.448** | .457 | .457 |
| ETTh2 | .382 | .378 | .381 | **.376** | .383 | .381 | .383 | **.380** |
| ETTm1 | .396 | **.390** | .396 | .391 | .398 | .400 | .397 | **.396** |
| ETTm2 | .284 | .279 | .284 | **.281** | .290 | **.283** | .288 | .286 |
| PEMS03 | .137 | .126 | **.121** | **.121** | .133 | .120 | .130 | **.119** |
| PEMS04 | .107 | .111 | **.095** | .099 | .096 | **.092** | .103 | .093 |
| PEMS07 | .091 | .091 | .090 | **.088** | .090 | .086 | .089 | **.085** |
| PEMS08 | .162 | **.139** | .144 | .142 | .157 | **.136** | .157 | .138 |
| Exchange | .363 | .361 | .361 | **.358** | .364 | .363 | .378 | **.361** |
| Weather | .257 | .256 | .256 | .256 | .252 | **.249** | .251 | .250 |
| Solar | .242 | .231 | .231 | **.230** | .244 | **.230** | .239 | .233 |
| ECL | .169 | .172 | .169 | **.168** | .174 | .175 | .174 | **.170** |
| Traffic | .412 | .410 | .410 | **.402** | .417 | .450 | .415 | **.414** |
| Average | .265 | .260 | .259 | **.257** | .266 | .263 | .266 | **.260** |

Table 9: Comparison of various SSL pretraining tasks.

| $H$ | SOR-Mamba | S-Mamba |
|---|---|---|
| 96 | $.378_{\pm.0003}$ | $.386_{\pm.0010}$ |
| 192 | $.428_{\pm.0002}$ | $.440_{\pm.0033}$ |
| 336 | $.464_{\pm.0002}$ | $.484_{\pm.0046}$ |
| 720 | $.464_{\pm.0004}$ | $.502_{\pm.0057}$ |

Table 10: Robustness to channel order.

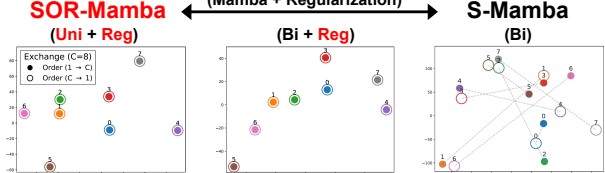

Figure 7: t-SNE of channel representations.

| Architecture for TD | ETT | | | | PEMS | | | | Exchange | Weather | Solar | ECL | Traffic | Avg. |
|---|---|---|---|---|---|---|---|---|---|---|---|---|---|---|
| | h1 | h2 | m1 | m2 | 03 | 04 | 07 | 08 | | | | | | |
| - | .446 | .386 | .397 | .286 | .139 | .109 | .096 | .164 | **.363** | .258 | .244 | .170 | .433 | .268 |
| Mamba | .447 | .386 | .398 | .285 | .140 | .109 | .097 | .165 | **.363** | .259 | .245 | .171 | .437 | .269 |
| MLP | **.442** | **.382** | **.396** | **.284** | **.137** | **.107** | **.091** | **.162** | **.363** | **.257** | **.242** | **.169** | **.412** | **.265** |

Table 11: Various architectures for capturing TD.

Furthermore, as CCM is designed to effectively capture CD in datasets, we compare the performance gain from three pretraining tasks based on the number of channels, with six datasets containing fewer than 100 channels and seven datasets containing 100 or more channels. Figure 6 shows the average performance gain from fine-tuning with the three tasks compared to SL, indicating that reconstruction is advantageous with fewer channels and masked modeling excels with more channels, while CCM consistently outperforms in both cases.

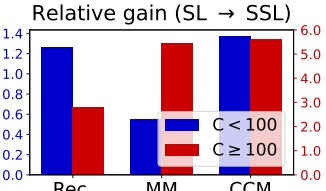

Figure 6: Comparison of SSL.

**Robustness to channel order.** To demonstrate that our method effectively addresses a sequential order bias, we conduct two analyses to show its robustness to the channel order. First, we evaluate the performance variations with five random permutations of channel order using ETTh1, where our method achieves a smaller standard deviation compared to S-Mamba, as shown in Table 10. Additional results with different datasets are described in Appendix H. Second, we visualize the output tokens of the encoder (i.e., embedding vectors of each channel) using t-SNE (Van der Maaten & Hinton, 2008) with Exchange. Figure 7 illustrates the results, showing that the tokens from the two views with reversed orders are consistent with regularization, while remaining inconsistent without it.

**Various architectures for TD.** Following the recent studies (Liu et al., 2024a; Wang et al., 2024) that suggest employing simple models, e.g., MLPs, to capture TD in TS, we utilize an MLP for this purpose. To examine the impact of different design choices of architecture for capturing TD, we consider two alternatives: 1) without employing any encoder for TD, and 2) using Mamba, following the previous work (Wang et al., 2024). Table 11 shows the results, demonstrating that our method is robust to the choice of encoder for TD, achieving the best performance with an MLP.

**Correlation in the data space and the latent space.** To demonstrate that CCM effectively preserves the relationships between channels from the data space to the latent space, we visualize the correlation matrices in both spaces with SOR-Mamba pretrained with CCM. Figure 8a shows the results on

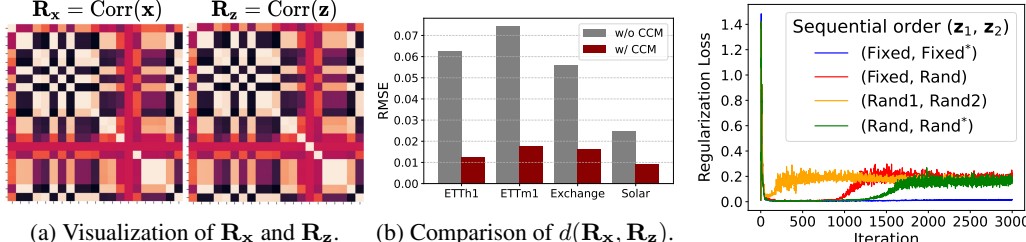

(a) Visualization of $\mathbf{R_x}$ and $\mathbf{R_z}$.   (b) Comparison of $d(\mathbf{R_x}, \mathbf{R_z})$.

Figure 8: Correlation matrices in data space and latent space.

Figure 9: Regularization loss.

| F: Fixed , R: Random , X⋆: Reverse of X | | | | | | Impr. (Robust.) |
|---|---|---|---|---|---|---|
| Order $\mathbf{z_1}$ | | $F$ | $F$ | $R_1$ | $R$ | |
| $\mathbf{z_2}$ | | $F^\star$ | $R$ | $R_2$ | $R^\star$ | |
| Dataset | $C$ | (a) | (b) | (c) | (d) | (d) → (a) |
| ETTh1 | 7 | **.442** | .443 | .446 | .443 | 0.2% |
| ETTh2 | 7 | **.382** | **.382** | .382 | **.382** | 0.0% |
| ETTm1 | 7 | **.396** | **.396** | .396 | **.396** | 0.0% |
| ETTm2 | 7 | **.284** | .285 | .285 | .285 | 0.4% |
| Exchange | 8 | **.363** | .364 | .365 | .364 | 0.3% |
| Weather | 21 | **.257** | .258 | .260 | .260 | 1.2% |
| Average | | **.354** | .355 | .356 | .355 | 0.3% |
| Solar | 137 | **.242** | .245 | .245 | .246 | 1.6% |
| PEMS03 | 358 | **.137** | .144 | .150 | .151 | 9.3% |
| PEMS04 | 307 | **.107** | .112 | .116 | .117 | 8.5% |
| PEMS07 | 883 | **.091** | .096 | .097 | .096 | 5.2% |
| PEMS08 | 170 | **.162** | .163 | .169 | .172 | 5.8% |
| ECL | 321 | **.169** | .174 | .181 | .183 | 7.7% |
| Traffic | 862 | **.412** | .422 | .423 | .423 | 2.6% |
| Average | | **.189** | .194 | .197 | .198 | 4.9% |

Table 12: Channel orders for two views.

| Dataset: Traffic ($L = 96, H = 96$) | (a) iTrans. | (b) S-Mamba | (c) SOR-Mamba | (b) → (c) Impr. |
|---|---|---|---|---|
| **# Parameters** | | | | |
| In projector | 0.05M | 0.05M | 0.05M | - |
| Encoder-CD | 4.20M | 6.97M | **3.48M** | **50.1%** |
| Encoder-TD | 2.11M | 2.11M | 2.11M | - |
| Out projector | 0.05M | 0.05M | 0.05M | - |
| Total | 6.52M | 9.29M | **5.80M** | **38.1%** |
| **Memory** | | | | |
| Complexity | $\mathcal{O}(C^2)$ | $\mathcal{O}(C)$ | $\mathcal{O}(C)$ | - |
| GPU memory (GB) | 1.36 | 0.33 | **0.32** | **4.2%** |
| **Computational time** | | | | |
| Train (sec.) | 115.5 | 108.3 | **102.1** | **5.7%** |
| Inference (ms) | 14.6 | 9.9 | **8.7** | **11.3%** |
| Avg. MSE (four $H$) | 0.428 | 0.417 | **0.402** | **3.6%** |

Table 13: Efficiency analysis.

the Weather dataset, which indicate that the relationships are effectively preserved with CCM. Additionally, we compare the distances between the matrices in both spaces, comparing SOR-Mamba without pretraining to the one pretrained with CCM. The results, illustrated in Figure 8b, show that the model pretrained with CCM exhibits a smaller difference between the matrices.

**Fixed vs. random order.** To generate two embedding vectors for regularization, we explore four candidates based on whether the channel order of $\mathbf{z_1}$ and $\mathbf{z_2}$ are fixed or randomly permuted in each iteration. Table 12 shows the results with the average MSE across four horizons, indicating that fixing the order yields better performance than permuting the order, especially with a large number of channels ($C \geq 100$). We argue that fixing the order leads to stable training, while permuting the order results in instability, as shown in the regularization loss curves for PEMS08 in Figure 9. Further analysis regarding the channel order is discussed in Appendix G.

**Efficiency analysis.** To demonstrate the efficiency of SOR-Mamba, we compare it with iTransformer and S-Mamba in terms of 1) the number of parameters, 2) memory usage, and 3) computational time. Table 13 shows the results, indicating that SOR-Mamba outperforms these methods in all three aspects, particularly reducing the number of parameters by up to 38.1% compared to S-Mamba. Note that the training time is measured per epoch, while the inference time is measured per data instance.

**Robustness to missingness.** To assess the robustness of our method in the presence of missing TS values, we conduct experiments in scenarios where 25%, 50%, and 75% of the TS values are randomly missing and interpolated using adjacent values. Figure 10 shows the average MSE across four horizons, indicating that our method remains robust even with significant amounts of missing data and that our method trained with missing values outperforms S-Mamba trained without missingness.

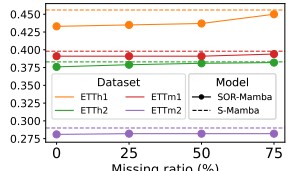

Figure 10: Missingness.

# 7 CONCLUSION

In this work, we introduce SOR-Mamba, a TS forecasting method that addresses the sequential order bias by incorporating a regularization strategy and removing the 1D-conv from Mamba. Additionally, we propose a novel pretraining task, CCM, to improve the model's ability to capture CD. Our results demonstrate that the proposed method is robust to variations in channel order, leading to superior performance and greater efficiency in both standard and transfer learning scenarios. We hope that our work motivates further research on sequential order-robust Mamba in domains where a sequential order is not inherent, such as in tabular data.

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

## A    DATASET STATISTICS AND EXPERIMENTAL SETUPS

**Dataset statistics.** We assess the performance of SOR-Mamba across 13 datasets, with the dataset statistics detailed in Table A.1, where $C$ and $T$ denote the number of channels and timesteps, respectively.

**Experimental setups.** We follow the same data processing steps and train-validation-test split protocol as used in S-Mamba (Wang et al., 2024), maintaining a chronological order in the separation of training, validation, and test sets, using a 6:2:2 ratio for the Solar-Energy, ETT, and PEMS datasets, and a 7:1:2 ratio for the other datasets. The results are shown in Table A.1, where $N$,$L$, and $H$ represent the dataset size, the size of the lookback window, and the size of the forecast horizon, respectively. For all datasets and all models, $L$ is uniformly set to 96. We do not tune any hyperparameters and adhere to those used in S-Mamba, except for $\lambda$, which is related to the proposed regularization, and is tuned using a grid search over $[0.001, 0.01, 0.1]$.

| Dataset | Statistics | | Experimental Setups | | |
|---|---|---|---|---|---|
| | $C$ | $T$ | $(N_{\text{train}}, N_{\text{val}}, N_{\text{test}})$ | $L$ | $H$ |
| ETTh1 (Zhou et al., 2021) | | 17420 | (8545, 2881, 2881) | | |
| ETTh2 (Zhou et al., 2021) | | 17420 | (8545, 2881, 2881) | | |
| ETTm1 (Zhou et al., 2021) | 7 | 69680 | (34465, 11521, 11521) | | |
| ETTm2 (Zhou et al., 2021) | | 69680 | (34465, 11521, 11521) | | |
| Exchange (Wu et al., 2021) | 8 | 7588 | (5120, 665, 1422) | | $\{96, 192, 336, 720\}$ |
| Weather (Wu et al., 2021) | 21 | 52696 | (36792, 5271, 10540) | | |
| ECL (Wu et al., 2021) | 321 | 26304 | (18317, 2633, 5261) | 96 | |
| Traffic (Wu et al., 2021) | 862 | 17544 | (12185, 1757, 3509) | | |
| Solar-Energy (Lai et al., 2018) | 137 | 52560 | (36601, 5161, 10417) | | |
| PEMS03 (Liu et al., 2022) | 358 | 26209 | (15617, 5135, 5135) | | |
| PEMS04 (Liu et al., 2022) | 307 | 15992 | (10172, 3375, 3375) | | $\{12, 24, 48, 96\}$ |
| PEMS07 (Liu et al., 2022) | 883 | 28224 | (16911, 5622, 5622) | | |
| PEMS08 (Liu et al., 2022) | 170 | 17856 | (10690, 3548, 3548) | | |

Table A.1: Datasets for TS forecasting.

## B    BASELINE METHODS

- S-Mamba (Wang et al., 2024): S-Mamba utilizes the bidirectional Mamba to capture channel dependencies in TS by scanning the channels from both directions.

- PatchTST (Nie et al., 2023): PatchTST segments TS into patches and feeds them into a Transformer in a channel independent manner.

- iTransformer (Liu et al., 2024a): iTransformer reverses the conventional role of the Transformer in the TS domain by treating each channel rather than patches as a token, thereby emphasizing channel dependencies over temporal dependencies.

- Crossformer (Zhang & Yan, 2023): Crossformer employs a cross-attention mechanism to capture both temporal and channel dependencies in TS.

- TimesNet (Wu et al., 2023): TimesNet captures both intraperiod and interperiod variations in 2D space using a parameter-efficient inception block.

- RLinear (Li et al., 2023): RLinear is a simple linear model that integrates reversible normalization and channel independence.

- DLinear (Zeng et al., 2023): DLinear is a simple linear model with channel independent architecture, that employs TS decomposition.

## C  S-MAMBA VS. SOR-MAMBA

Figure C.1 visualizes the comparison between S-Mamba (Wang et al., 2024), which employs the bidirectional Mamba to capture CD, and our method, SOR-Mamba, which uses a single unidirectional Mamba with regularization to capture CD.

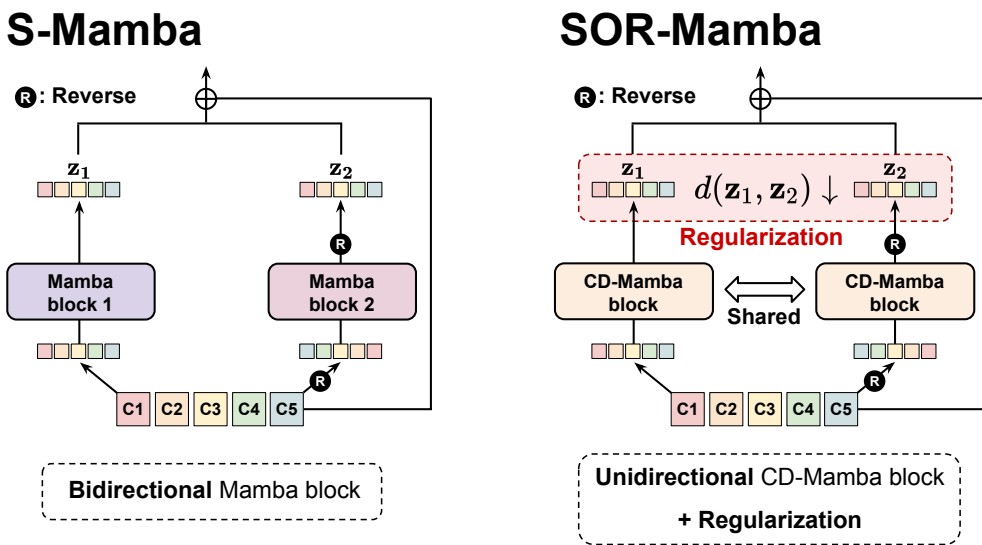

Figure C.1: Comparison of S-Mamba and SOR-Mamba.

## D REMOVAL OF 1D-CONVOLUTION

The original Mamba block (Gu & Dao, 2023) integrates the H3 block (Fu et al., 2023) with a gated MLP, where the H3 block uses a 1D-conv before the SSM layer to capture local information within nearby tokens, as illustrated in Figure D.1. However, since channels in TS do not have an inherent sequential order, we eliminate the 1D-conv from the Mamba block, resulting in the proposed CD-Mamba block. Figure D.2 shows the overall architecture of the proposed CD-Mamba block, where the 1D-conv before the selective SSM is removed from the original Mamba block (Gu & Dao, 2023).

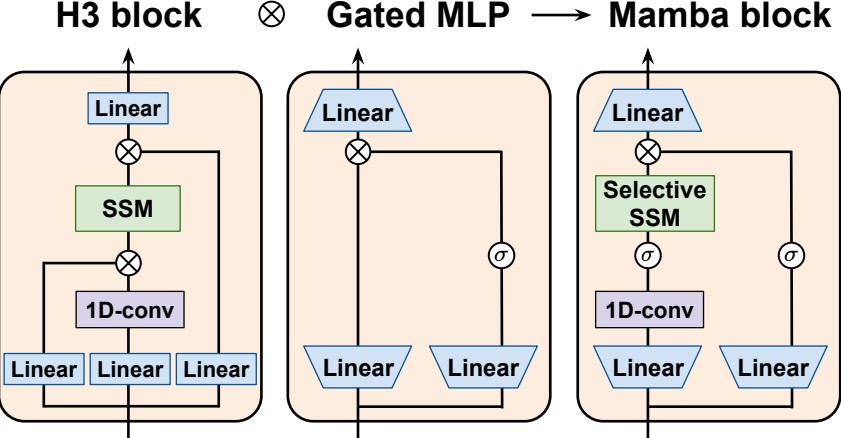

Figure D.1: **Architecture of the original Mamba block.** The original Mamba block contains 1D-conv before the SSM layer to capture local information within nearby tokens.1

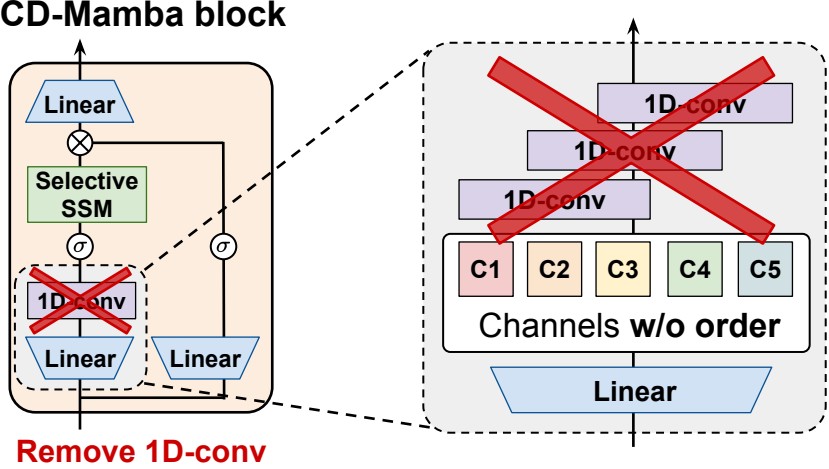

Figure D.2: **Architecture of the CD-Mamba block.** 1D-conv before the selective SSM is removed from the original Mamba block, as the channels do not have a sequential order.

# E  FULL RESULTS OF TIME SERIES FORECASTING

Table E.1 shows the full results of TS forecasting tasks across four different horizons, highlighting the effectiveness of our method.

| Models | | SOR-Mamba FT | | SOR-Mamba SL | | S-Mamba | | iTransformer | | RLinear | | PatchTST | | Crossformer | | TiDE | | TimesNet | | DLinear | |
|---|---|---|---|---|---|---|---|---|---|---|---|---|---|---|---|---|---|---|---|---|---|
| Metric | | MSE | MAE | MSE | MAE | MSE | MAE | MSE | MAE | MSE | MAE | MSE | MAE | MSE | MAE | MSE | MAE | MSE | MAE | MSE | MAE |
| ETTh1 | 96 | .377 | .398 | .385 | .398 | .385 | .404 | .387 | .405 | .386 | .395 | .414 | .419 | .423 | .448 | .479 | .464 | .384 | .402 | .386 | .400 |
| | 192 | .428 | .429 | .435 | .428 | .445 | .441 | .441 | .436 | .437 | .424 | .460 | .445 | .471 | .474 | .525 | .492 | .436 | .429 | .437 | .432 |
| | 336 | .464 | .448 | .474 | .448 | .491 | .462 | .487 | .458 | .479 | .446 | .501 | .466 | .570 | .546 | .565 | .515 | .491 | .469 | .481 | .459 |
| | 720 | .464 | .469 | .478 | .471 | .506 | .497 | .509 | .494 | .481 | .470 | .500 | .488 | .653 | .621 | .594 | .558 | .521 | .500 | .519 | .516 |
| | Avg. | .433 | .436 | .442 | .438 | .457 | .452 | .457 | .449 | .446 | .434 | .469 | .454 | .529 | .522 | .541 | .507 | .458 | .450 | .456 | .452 |
| ETTh2 | 96 | .292 | .348 | .299 | .348 | .297 | .349 | .301 | .350 | .288 | .338 | .302 | .348 | .745 | .584 | .400 | .440 | .340 | .374 | .333 | .387 |
| | 192 | .372 | .397 | .375 | .399 | .378 | .399 | .381 | .399 | .374 | .390 | .388 | .400 | .877 | .656 | .528 | .509 | .402 | .414 | .477 | .476 |
| | 336 | .415 | .431 | .423 | .435 | .425 | .435 | .427 | .434 | .415 | .426 | .426 | .433 | 1.043 | .731 | .643 | .571 | .452 | .452 | .594 | .541 |
| | 720 | .423 | .445 | .431 | .446 | .432 | .448 | .430 | .446 | .420 | .440 | .431 | .446 | 1.104 | .763 | .874 | .679 | .462 | .468 | .831 | .657 |
| | Avg. | .376 | .405 | .382 | .407 | .383 | .408 | .384 | .407 | .374 | .398 | .387 | .407 | .942 | .684 | .611 | .550 | .414 | .427 | .559 | .515 |
| ETTm1 | 96 | .324 | .362 | .326 | .367 | .326 | .368 | .342 | .377 | .355 | .376 | .329 | .367 | .404 | .426 | .364 | .387 | .338 | .375 | .345 | .372 |
| | 192 | .369 | .385 | .375 | .387 | .378 | .393 | .383 | .396 | .391 | .392 | .367 | .385 | .450 | .451 | .398 | .404 | .374 | .387 | .380 | .389 |
| | 336 | .402 | .408 | .408 | .408 | .410 | .414 | .418 | .418 | .424 | .415 | .399 | .410 | .532 | .515 | .428 | .425 | .410 | .411 | .413 | .413 |
| | 720 | .467 | .444 | .472 | .444 | .474 | .451 | .487 | .456 | .487 | .450 | .454 | .439 | .666 | .589 | .487 | .461 | .478 | .450 | .474 | .453 |
| | Avg. | .391 | .400 | .396 | .401 | .398 | .407 | .408 | .412 | .414 | .407 | .387 | .400 | .513 | .496 | .419 | .419 | .400 | .406 | .403 | .407 |
| ETTm2 | 96 | .179 | .261 | .181 | .265 | .182 | .266 | .186 | .272 | .182 | .265 | .175 | .259 | .287 | .366 | .207 | .305 | .187 | .267 | .193 | .292 |
| | 192 | .241 | .304 | .246 | .307 | .252 | .313 | .254 | .314 | .246 | .304 | .241 | .302 | .414 | .492 | .290 | .364 | .249 | .309 | .284 | .362 |
| | 336 | .302 | .342 | .306 | .345 | .313 | .349 | .317 | .353 | .307 | .342 | .305 | .343 | .597 | .542 | .377 | .422 | .321 | .351 | .369 | .427 |
| | 720 | .401 | .400 | .403 | .401 | .416 | .409 | .412 | .407 | .407 | .398 | .402 | .400 | 1.730 | 1.042 | .558 | .524 | .408 | .403 | .554 | .522 |
| | Avg. | .281 | .327 | .284 | .329 | .290 | .333 | .293 | .337 | .286 | .327 | .281 | .326 | .757 | .610 | .358 | .404 | .291 | .333 | .350 | .401 |
| PEMS03 | 12 | .066 | .170 | .066 | .170 | .066 | .171 | .071 | .174 | .126 | .236 | .099 | .216 | .090 | .203 | .178 | .305 | .085 | .192 | .122 | .243 |
| | 24 | .088 | .197 | .090 | .200 | .088 | .197 | .097 | .208 | .246 | .334 | .142 | .259 | .121 | .240 | .257 | .371 | .118 | .223 | .201 | .317 |
| | 48 | .134 | .245 | .167 | .280 | .165 | .277 | .161 | .272 | .551 | .529 | .211 | .319 | .202 | .317 | .379 | .463 | .155 | .260 | .333 | .425 |
| | 96 | .193 | .297 | .225 | .318 | .213 | .313 | .240 | .338 | 1.057 | .787 | .269 | .370 | .262 | .367 | .490 | .539 | .228 | .317 | .457 | .515 |
| | Avg. | .121 | .227 | .137 | .242 | .133 | .240 | .142 | .248 | .495 | .471 | .180 | .291 | .169 | .281 | .326 | .419 | .147 | .248 | .278 | .375 |
| PEMS04 | 12 | .074 | .175 | .077 | .180 | .073 | .177 | .081 | .188 | .138 | .252 | .105 | .224 | .098 | .218 | .219 | .340 | .087 | .195 | .148 | .272 |
| | 24 | .086 | .192 | .091 | .197 | .084 | .192 | .099 | .211 | .258 | .348 | .153 | .275 | .131 | .256 | .292 | .398 | .103 | .215 | .224 | .340 |
| | 48 | .106 | .214 | .115 | .221 | .101 | .213 | .133 | .246 | .229 | .339 | .229 | .339 | .205 | .326 | .409 | .478 | .136 | .250 | .355 | .437 |
| | 96 | .129 | .233 | .143 | .248 | .125 | .236 | .172 | .283 | 1.137 | .820 | .291 | .389 | .402 | .457 | .492 | .532 | .190 | .303 | .452 | .504 |
| | Avg. | .099 | .203 | .107 | .212 | .096 | .205 | .121 | .232 | .526 | .491 | .195 | .307 | .209 | .314 | .353 | .437 | .129 | .241 | .295 | .388 |
| PEMS07 | 12 | .059 | .155 | .060 | .156 | .060 | .157 | .067 | .165 | .118 | .235 | .105 | .207 | .094 | .200 | .173 | .304 | .082 | .181 | .115 | .242 |
| | 24 | .076 | .174 | .082 | .182 | .082 | .184 | .088 | .190 | .242 | .341 | .150 | .262 | .139 | .247 | .271 | .383 | .101 | .204 | .210 | .329 |
| | 48 | .098 | .199 | .107 | .209 | .100 | .204 | .113 | .218 | .562 | .541 | .253 | .340 | .311 | .369 | .446 | .495 | .134 | .238 | .398 | .458 |
| | 96 | .117 | .218 | .117 | .218 | .117 | .218 | .172 | .283 | 1.096 | .795 | .346 | .404 | .396 | .442 | .628 | .577 | .181 | .279 | .594 | .553 |
| | Avg. | .088 | .186 | .091 | .191 | .090 | .191 | .102 | .205 | .504 | .478 | .211 | .303 | .235 | .315 | .380 | .440 | .124 | .225 | .329 | .395 |
| PEMS08 | 12 | .078 | .178 | .076 | .176 | .076 | .178 | .088 | .193 | .133 | .247 | .168 | .232 | .165 | .214 | .227 | .343 | .112 | .212 | .154 | .276 |
| | 24 | .103 | .205 | .109 | .212 | .110 | .216 | .138 | .243 | .249 | .343 | .224 | .281 | .215 | .260 | .318 | .409 | .141 | .248 | .248 | .353 |
| | 48 | .159 | .250 | .172 | .264 | .173 | .254 | .334 | .353 | .569 | .544 | .321 | .354 | .315 | .355 | .497 | .510 | .198 | .283 | .440 | .470 |
| | 96 | .229 | .295 | .290 | .334 | .271 | .321 | .458 | .436 | 1.166 | .814 | .408 | .417 | .377 | .397 | .721 | .592 | .320 | .351 | .674 | .565 |
| | Avg. | .142 | .232 | .162 | .247 | .157 | .242 | .254 | .306 | .529 | .487 | .268 | .321 | .268 | .307 | .441 | .464 | .193 | .271 | .379 | .416 |
| Exchange | 96 | .085 | .204 | .085 | .205 | .086 | .206 | .086 | .206 | .093 | .217 | .088 | .205 | .256 | .367 | .094 | .218 | .107 | .234 | .088 | .218 |
| | 192 | .179 | .301 | .179 | .301 | .181 | .303 | .177 | .299 | .184 | .307 | .176 | .299 | .470 | .509 | .184 | .307 | .226 | .344 | .176 | .315 |
| | 336 | .329 | .415 | .331 | .417 | .331 | .417 | .338 | .422 | .351 | .432 | .301 | .397 | 1.268 | .883 | .349 | .431 | .367 | .448 | .313 | .427 |
| | 720 | .838 | .690 | .860 | .698 | .858 | .599 | .847 | .691 | .886 | .714 | .901 | .714 | 1.767 | 1.068 | .852 | .698 | .964 | .746 | .839 | .695 |
| | Avg. | .358 | .402 | .363 | .405 | .364 | .407 | .368 | .409 | .378 | .417 | .367 | .404 | .940 | .707 | .370 | .413 | .416 | .443 | .354 | .414 |
| Weather | 96 | .174 | .212 | .175 | .215 | .165 | .209 | .174 | .215 | .192 | .232 | .177 | .218 | .158 | .230 | .202 | .261 | .172 | .220 | .196 | .255 |
| | 192 | .221 | .255 | .221 | .255 | .215 | .255 | .224 | .258 | .240 | .271 | .225 | .259 | .206 | .277 | .242 | .298 | .219 | .261 | .237 | .296 |
| | 336 | .277 | .295 | .277 | .296 | .273 | .296 | .281 | .298 | .292 | .307 | .278 | .297 | .273 | .335 | .287 | .335 | .280 | .306 | .283 | .335 |
| | 720 | .353 | .348 | .355 | .348 | .353 | .349 | .359 | .351 | .364 | .353 | .354 | .348 | .398 | .418 | .351 | .386 | .365 | .359 | .345 | .381 |
| | Avg. | .256 | .277 | .257 | .278 | .252 | .277 | .260 | .281 | .272 | .291 | .259 | .281 | .259 | .315 | .271 | .320 | .259 | .287 | .265 | .317 |
| Solar | 96 | .194 | .229 | .207 | .246 | .207 | .246 | .201 | .234 | .322 | .339 | .234 | .286 | .310 | .331 | .312 | .399 | .250 | .292 | .290 | .378 |
| | 192 | .228 | .256 | .239 | .270 | .240 | .272 | .238 | .261 | .359 | .356 | .267 | .310 | .734 | .725 | .339 | .416 | .296 | .318 | .320 | .398 |
| | 336 | .247 | .276 | .260 | .287 | .262 | .290 | .248 | .273 | .397 | .369 | .290 | .315 | .750 | .735 | .368 | .430 | .319 | .330 | .353 | .415 |
| | 720 | .251 | .275 | .264 | .291 | .267 | .293 | .249 | .275 | .397 | .356 | .289 | .317 | .769 | .765 | .370 | .425 | .338 | .337 | .356 | .413 |
| | Avg. | .230 | .259 | .242 | .274 | .244 | .275 | .234 | .261 | .369 | .356 | .270 | .307 | .641 | .639 | .347 | .417 | .301 | .319 | .330 | .401 |
| ECL | 96 | .139 | .235 | .139 | .233 | .139 | .237 | .148 | .240 | .201 | .281 | .181 | .270 | .219 | .314 | .237 | .329 | .168 | .272 | .197 | .282 |
| | 192 | .160 | .254 | .158 | .249 | .165 | .261 | .167 | .258 | .201 | .283 | .188 | .274 | .231 | .322 | .236 | .330 | .184 | .289 | .196 | .285 |
| | 336 | .176 | .271 | .177 | .271 | .177 | .272 | .179 | .274 | .215 | .298 | .204 | .293 | .246 | .337 | .249 | .344 | .198 | .300 | .209 | .301 |
| | 720 | .198 | .292 | .201 | .293 | .214 | .304 | .220 | .310 | .257 | .331 | .246 | .324 | .280 | .363 | .284 | .373 | .220 | .320 | .245 | .333 |
| | Avg. | .168 | .264 | .169 | .262 | .174 | .269 | .179 | .270 | .219 | .298 | .205 | .290 | .244 | .334 | .251 | .344 | .192 | .295 | .212 | .300 |
| Traffic | 96 | .378 | .261 | .378 | .259 | .379 | .260 | .395 | .268 | .649 | .389 | .462 | .295 | .522 | .290 | .805 | .493 | .593 | .321 | .650 | .396 |
| | 192 | .393 | .269 | .399 | .270 | .409 | .272 | .417 | .277 | .601 | .366 | .466 | .296 | .530 | .293 | .756 | .474 | .617 | .336 | .598 | .370 |
| | 336 | .399 | .272 | .416 | .279 | .418 | .277 | .433 | .283 | .609 | .369 | .482 | .304 | .558 | .305 | .762 | .477 | .629 | .336 | .605 | .373 |
| | 720 | .437 | .290 | .456 | .297 | .461 | .297 | .467 | .300 | .647 | .387 | .514 | .322 | .589 | .328 | .717 | .449 | .640 | .350 | .645 | .394 |
| | Avg. | .402 | .273 | .412 | .276 | .417 | .277 | .428 | .282 | .626 | .378 | .481 | .304 | .550 | .304 | .760 | .473 | .620 | .350 | .625 | .383 |
| 1st Count | | 33 | 31 | 7 | 10 | 10 | 7 | 1 | 3 | 3 | 9 | 8 | 7 | 3 | 0 | 0 | 0 | 0 | 0 | 2 | 0 |
| 2nd Count | | 15 | 19 | 18 | 19 | 13 | 13 | 9 | 6 | 2 | 2 | 1 | 6 | 0 | 0 | 0 | 0 | 0 | 1 | 2 | 0 |

Table E.1: Full results of TS forecasting tasks.

# F    ABLATION STUDY

To demonstrate the effectiveness of our method, we conduct an ablation study using four ETT datasets (Zhou et al., 2021) to assess the impact of the following components, where the results are shown in Table F.1. The results indicate that incorporating all components yields the best performance, and adding the regularization term enhances the performance even with the bidirectional Mamba.

| Method | Mamba | | Reg. | CCM | ETTh1 | ETTh2 | ETTm1 | ETTm2 | Avg. |
| | # | w/o conv. | | | | | | | |
| --- | --- | --- | --- | --- | --- | --- | --- | --- | --- |
| S-Mamba | Bi | - | - | - | .457 | .383 | .398 | .290 | .382 |
| - | Bi | ✓ | - | - | .441 | .383 | .396 | .285 | .376 |
| - | Bi | - | ✓ | | .452 | .382 | .394 | .286 | .378 |
| - | Bi | ✓ | ✓ | | .443 | .381 | .393 | .285 | .376 |
| - | Bi | ✓ | ✓ | ✓ | .435 | **.376** | **.390** | **.281** | **.370** |
| - | Uni | - | - | - | .455 | .383 | .403 | .289 | .383 |
| - | Uni | ✓ | - | - | .442 | .382 | .400 | .285 | .377 |
| - | Uni | - | ✓ | - | .449 | .382 | .396 | .285 | .378 |
| - | Uni | ✓ | ✓ | - | .442 | .382 | .396 | .284 | .376 |
| SOR-Mamba | Uni | ✓ | ✓ | ✓ | **.433** | **.376** | .391 | **.281** | **.370** |

Table F.1: Ablation studies with four ETT datasets.

# G    CHANNEL ORDERS FOR TWO VIEWS

Figure G.1 illustrates the four candidates for generating two embedding vectors, $z_1$ and $z_2$, for regularization, based on whether the channel order is fixed or randomly permuted in each iteration. Results in Table 12 indicate that fixing the order during training yields the best performance, with performance degrading as the order becomes random, especially with many channels, though it remains robust with fewer channels. We argue that a fixed order is preferable due to the instability introduced by randomness during training, as shown in Figure G.1, which displays the training loss for two datasets (Zhou et al., 2021; Liu et al., 2022) with varying numbers of channels. The figure indicates that a random order causes instability, particularly with the regularization loss.

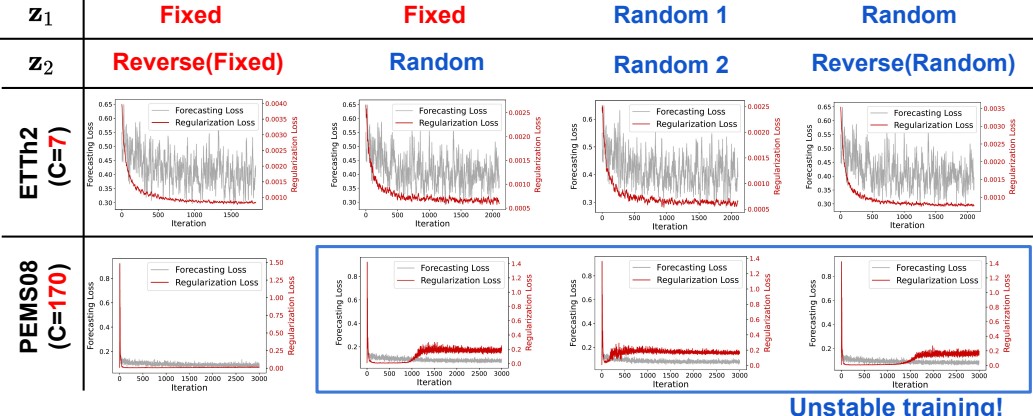

Figure G.1: Fixed vs. random order for generating two views, $z_1$ and $z_2$.

## H  ROBUSTNESS TO CHANNEL ORDER

To demonstrate that the proposed method effectively addresses the sequential order bias, we evaluate performance variations by permuting the channel order with five datasets (Zhou et al., 2021; Wu et al., 2021). Table H.1 shows the results, which indicate a small standard deviation across all horizons.

| $H$ | ETTh1 | ETTh2 | ETTm1 | ETTm2 | Exchange |
|---|---|---|---|---|---|
| 96 | $.377_{\pm.0003}$ | $.292_{\pm.0011}$ | $.324_{\pm.0005}$ | $.179_{\pm.0003}$ | $.085_{\pm.0001}$ |
| 192 | $.428_{\pm.0002}$ | $.372_{\pm.0000}$ | $.369_{\pm.0005}$ | $.241_{\pm.0002}$ | $.179_{\pm.0001}$ |
| 336 | $.464_{\pm.0002}$ | $.415_{\pm.0002}$ | $.402_{\pm.0003}$ | $.302_{\pm.0001}$ | $.329_{\pm.0002}$ |
| 720 | $.464_{\pm.0004}$ | $.423_{\pm.0001}$ | $.467_{\pm.0009}$ | $.401_{\pm.0001}$ | $.838_{\pm.0014}$ |
| Avg. | $.434_{\pm.0002}$ | $.423_{\pm.0003}$ | $.391_{\pm.0001}$ | $.281_{\pm.0001}$ | $.358_{\pm.0003}$ |

Table H.1: Robustness to channel order.

## I  ROBUSTNESS TO HYPERPARAMETER $\lambda$

Table I.1 shows the average MSE across four different horizons for the four ETT datasets (Zhou et al., 2021), using various values of $\lambda$ that control the contribution of the regularization term. The results demonstrate the effectiveness of the regularization and its robustness to $\lambda$.

| Dataset | SOR-Mamba w/o Reg. | SOR-Mamba w/ Reg. | | | | S-Mamba |
|---|---|---|---|---|---|---|
| | 0 | 0.001 | 0.01 | 0.1 | 0.2 | |
| ETTh1 | .439 | **.433** | **.433** | **.433** | **.433** | .457 |
| ETTh2 | .382 | **.376** | **.376** | **.376** | **.376** | .383 |
| ETTm1 | .403 | **.391** | **.391** | **.391** | **.391** | .398 |
| ETTm2 | .285 | **.281** | **.281** | **.281** | **.281** | .290 |

Table I.1: Robustness to choice of $\lambda$ for regularization.

## J  PSEUDOCODE OF CCM

Algorithm 2 shows the pseudocode for the proposed pretraining task, channel correlation modeling (CCM), where an arbitrary TS encoder can be employed.

---
**Algorithm 2** Channel Correlation Modeling (CCM)

**Input**: $\mathbf{X} = [\mathbf{X}_1, \ldots, \mathbf{X}_L] : (B, L, C)$

1: $\mathbf{R_X} : (B, C, C) \leftarrow$ Calculate correlation matrix with $\mathbf{X}$
2: $\mathbf{Z} : (B, C, D) \leftarrow$ Encoder($\mathbf{X}$)
3: $\mathbf{R_Z} : (B, C, C) \leftarrow$ Calculate correlation matrix with $\mathbf{Z}$
4: Minimize $d(\mathbf{R_X}, \mathbf{R_Z})$

---

## K  ROBUSTNESS TO DISTANCE METRIC

To assess whether SOR-Mamba is sensitive to the choice of distance metric $d$ for the regularization term and CCM when comparing the two matrices, we compare various metrics, including (negative) cosine similarity, $\ell_1$ loss, and $\ell_2$ loss. Tables K.1 and K.2 show the average MSE across four different horizons for the distance metric used in the regularization term and CCM, respectively, demonstrating that the performance is robust to the choice of distance metric, where we choose $\ell_2$ loss throughout the experiment for both metrics.

| Dataset | SOR-Mamba-SL | | | S-Mamba |
|---|---|---|---|---|
| | Cosine | $\ell_1$ Loss | $\ell_2$ Loss | |
| ETTh1 | **.442** | **.442** | **.442** | .457 |
| ETTh2 | **.382** | **.382** | **.382** | .383 |
| ETTm1 | **.396** | **.396** | **.396** | .398 |
| ETTm2 | **.284** | **.284** | **.284** | .290 |
| PEMS03 | .145 | .147 | .137 | **.133** |
| PEMS04 | .105 | .105 | .107 | **.096** |
| PEMS07 | .091 | .091 | .091 | **.090** |
| PEMS08 | .162 | .159 | .162 | **.157** |
| Exchange | .365 | .365 | **.363** | .364 |
| Weather | .256 | .257 | .257 | **.252** |
| Solar | **.242** | **.242** | **.242** | .244 |
| ECL | **.167** | .168 | .169 | .174 |
| Traffic | .414 | .414 | **.412** | .417 |
| Average | **.265** | **.265** | **.265** | .266 |

Table K.1: Robustness to $d$ for regularization.

| Dataset | SOR-Mamba-SSL | | S-Mamba |
|---|---|---|---|
| | $\ell_1$ Loss | $\ell_2$ Loss | |
| ETTh1 | .434 | **.433** | .457 |
| ETTh2 | .379 | **.376** | .383 |
| ETTm1 | **.391** | **.391** | .398 |
| ETTm2 | **.281** | **.281** | .290 |
| PEMS03 | **.121** | **.121** | .133 |
| PEMS04 | .099 | .099 | **.096** |
| PEMS07 | .089 | **.088** | .090 |
| PEMS08 | **.140** | .142 | .157 |
| Exchange | **.358** | **.358** | .364 |
| Weather | .256 | .256 | **.252** |
| Solar | .232 | **.230** | .244 |
| ECL | **.167** | .168 | 174 |
| Traffic | **.402** | **.402** | .417 |
| Average | .258 | **.257** | .266 |

Table K.2: Robustness to $d$ for CCM.

## L  SIZE OF LOOKBACK WINDOW VS. PERFORMANCE

Following the previous works (Liu et al., 2024a; Wang et al., 2024), we conduct an experiment to evaluate the performance as the size of the lookback window ($L$) increases, using four datasets: ECL (Wu et al., 2021), Traffic (Wu et al., 2021), PEMS04 (Liu et al., 2022), and ETTm1 (Zhou et al., 2021), with the baseline methods and results from S-Mamba (Wang et al., 2024). The results, shown in Figure L.1, indicate that the performance remains robust to the choice of $L$ for some datasets and even improves with larger $L$ for others.

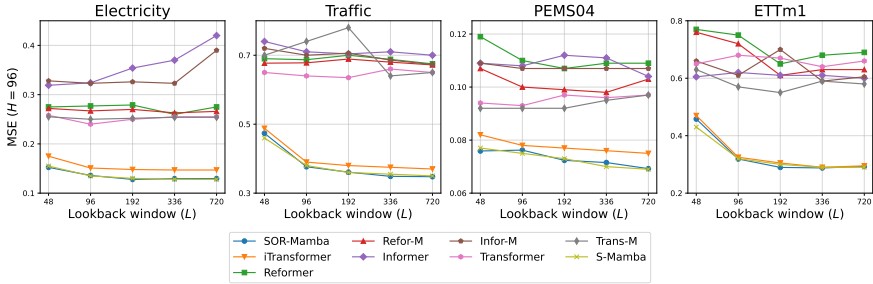

Figure L.1: **Size of lookback window vs. performance.** Forecasting performance on four datasets with the lookback length $L \in \{48, 96, 192, 336, 720\}$, with forecast horizon $H = 12$ for PEMS04 and $H = 96$ for other datasets.

## M    COMPARISON OF GPU MEMORY USAGE

Figure M.1 visualizes GPU memory usage by dataset and method, demonstrating that our method is more efficient than both S-Mamba (Wang et al., 2024) and iTransformer (Liu et al., 2024a). Specifically, Mamba-based methods are more efficient than Transformer-based methods when $C$ is large, as Mamba has nearly-linear complexity, whereas Transformers have quadratic complexity.

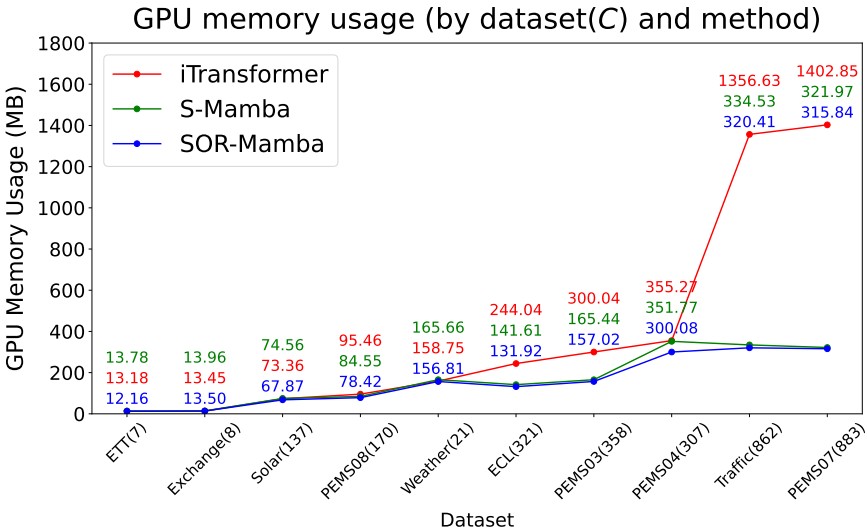

Figure M.1: Comparison of GPU memory usage.

# N STATISTICS OF RESULTS OVER MULTIPLE RUNS

To assess the consistency of SOR-Mamba's performance, we present the statistics from results using five different random seeds. We calculate the mean and standard deviation for both MSE and MAE, detailed in Tables N.1, N.2, and N.3. which reveal that our method maintains consistent performance in both self-supervised and supervised settings.

| Models | | Ours | | | |
|---|---|---|---|---|---|
| | | FT | | SL | |
| Metric | | MSE | MAE | MSE | MAE |
| ETTh1 | 96 | $.377_{\pm.001}$ | $.398_{\pm.001}$ | $.385_{\pm.000}$ | $.398_{\pm.000}$ |
| | 192 | $.428_{\pm.001}$ | $.429_{\pm.000}$ | $.432_{\pm.001}$ | $.428_{\pm.000}$ |
| | 336 | $.464_{\pm.001}$ | $.448_{\pm.001}$ | $.476_{\pm.000}$ | $.448_{\pm.000}$ |
| | 720 | $.464_{\pm.001}$ | $.469_{\pm.006}$ | $.476_{\pm.003}$ | $.476_{\pm.002}$ |
| | Avg. | $.433_{\pm.000}$ | $.436_{\pm.002}$ | $.442_{\pm.001}$ | $.438_{\pm.000}$ |
| ETTh2 | 96 | $.292_{\pm.004}$ | $.348_{\pm.003}$ | $.299_{\pm.001}$ | $.348_{\pm.001}$ |
| | 192 | $.372_{\pm.001}$ | $.397_{\pm.001}$ | $.375_{\pm.001}$ | $.399_{\pm.001}$ |
| | 336 | $.415_{\pm.001}$ | $.431_{\pm.000}$ | $.423_{\pm.000}$ | $.435_{\pm.000}$ |
| | 720 | $.423_{\pm.001}$ | $.445_{\pm.001}$ | $.431_{\pm.002}$ | $.446_{\pm.001}$ |
| | Avg. | $.376_{\pm.001}$ | $.405_{\pm.001}$ | $.382_{\pm.001}$ | $.407_{\pm.000}$ |
| ETTm1 | 96 | $.324_{\pm.002}$ | $.362_{\pm.002}$ | $.324_{\pm.004}$ | $.367_{\pm.003}$ |
| | 192 | $.369_{\pm.002}$ | $.385_{\pm.001}$ | $.375_{\pm.002}$ | $.387_{\pm.001}$ |
| | 336 | $.402_{\pm.002}$ | $.408_{\pm.001}$ | $.408_{\pm.000}$ | $.408_{\pm.000}$ |
| | 720 | $.467_{\pm.002}$ | $.444_{\pm.001}$ | $.472_{\pm.001}$ | $.444_{\pm.001}$ |
| | Avg. | $.391_{\pm.001}$ | $.400_{\pm.001}$ | $.396_{\pm.001}$ | $.401_{\pm.001}$ |
| ETTm2 | 96 | $.179_{\pm.001}$ | $.261_{\pm.001}$ | $.181_{\pm.000}$ | $.265_{\pm.000}$ |
| | 192 | $.241_{\pm.000}$ | $.304_{\pm.000}$ | $.246_{\pm.001}$ | $.307_{\pm.001}$ |
| | 336 | $.302_{\pm.002}$ | $.342_{\pm.002}$ | $.306_{\pm.001}$ | $.345_{\pm.000}$ |
| | 720 | $.401_{\pm.002}$ | $.400_{\pm.002}$ | $.403_{\pm.002}$ | $.401_{\pm.001}$ |
| | Avg. | $.281_{\pm.001}$ | $.327_{\pm.000}$ | $.284_{\pm.001}$ | $.329_{\pm.000}$ |

Table N.1: Results of TS forecasting over five runs - 1) ETT datasets.

| Models | | Ours | | | |
|---|---|---|---|---|---|
| | | FT | | SL | |
| Metric | | MSE | MAE | MSE | MAE |
| PEMS03 | 12 | $.066_{\pm.001}$ | $.170_{\pm.001}$ | $.066_{\pm.001}$ | $.170_{\pm.001}$ |
| | 24 | $.088_{\pm.001}$ | $.197_{\pm.001}$ | $.090_{\pm.001}$ | $.200_{\pm.001}$ |
| | 48 | $.134_{\pm.002}$ | $.245_{\pm.003}$ | $.167_{\pm.001}$ | $.280_{\pm.001}$ |
| | 96 | $.193_{\pm.005}$ | $.297_{\pm.006}$ | $.225_{\pm.003}$ | $.318_{\pm.002}$ |
| | Avg. | $.121_{\pm.002}$ | $.227_{\pm.002}$ | $.137_{\pm.001}$ | $.242_{\pm.001}$ |
| PEMS04 | 12 | $.074_{\pm.002}$ | $.175_{\pm.003}$ | $.077_{\pm.000}$ | $.180_{\pm.000}$ |
| | 24 | $.086_{\pm.003}$ | $.192_{\pm.005}$ | $.091_{\pm.001}$ | $.197_{\pm.001}$ |
| | 48 | $.106_{\pm.001}$ | $.214_{\pm.005}$ | $.115_{\pm.002}$ | $.221_{\pm.003}$ |
| | 96 | $.129_{\pm.003}$ | $.233_{\pm.004}$ | $.143_{\pm.002}$ | $.248_{\pm.002}$ |
| | Avg. | $.099_{\pm.001}$ | $.203_{\pm.002}$ | $.107_{\pm.001}$ | $.212_{\pm.001}$ |
| PEMS07 | 12 | $.059_{\pm.001}$ | $.155_{\pm.001}$ | $.060_{\pm.000}$ | $.156_{\pm.000}$ |
| | 24 | $.076_{\pm.005}$ | $.174_{\pm.004}$ | $.082_{\pm.000}$ | $.182_{\pm.000}$ |
| | 48 | $.098_{\pm.001}$ | $.199_{\pm.001}$ | $.107_{\pm.001}$ | $.209_{\pm.000}$ |
| | 96 | $.117_{\pm.003}$ | $.218_{\pm.003}$ | $.117_{\pm.001}$ | $.218_{\pm.001}$ |
| | Avg. | $.088_{\pm.001}$ | $.186_{\pm.001}$ | $.091_{\pm.000}$ | $.191_{\pm.000}$ |
| PEMS08 | 12 | $.078_{\pm.000}$ | $.178_{\pm.000}$ | $.076_{\pm.001}$ | $.176_{\pm.000}$ |
| | 24 | $.103_{\pm.001}$ | $.205_{\pm.002}$ | $.109_{\pm.001}$ | $.212_{\pm.001}$ |
| | 48 | $.159_{\pm.001}$ | $.250_{\pm.001}$ | $.172_{\pm.003}$ | $.264_{\pm.003}$ |
| | 96 | $.229_{\pm.001}$ | $.295_{\pm.002}$ | $.290_{\pm.002}$ | $.334_{\pm.002}$ |
| | Avg. | $.142_{\pm.000}$ | $.232_{\pm.001}$ | $.162_{\pm.001}$ | $.247_{\pm.001}$ |

Table N.2: Results of TS forecasting over five runs - 2) PEMS datasets.

| Models | | Ours | | | |
|---|---|---|---|---|---|
| | | FT | | SL | |
| Metric | | MSE | MAE | MSE | MAE |
| Exchange | 96 | $.085_{\pm.001}$ | $.204_{\pm.002}$ | $.085_{\pm.001}$ | $.205_{\pm.001}$ |
| | 192 | $.179_{\pm.000}$ | $.301_{\pm.000}$ | $.179_{\pm.002}$ | $.301_{\pm.001}$ |
| | 336 | $.329_{\pm.001}$ | $.415_{\pm.001}$ | $.331_{\pm.000}$ | $.417_{\pm.000}$ |
| | 720 | $.838_{\pm.005}$ | $.690_{\pm.002}$ | $.860_{\pm.001}$ | $.698_{\pm.001}$ |
| | Avg. | $.358_{\pm.001}$ | $.402_{\pm.001}$ | $.363_{\pm.001}$ | $.405_{\pm.001}$ |
| Weather | 96 | $.174_{\pm.000}$ | $.212_{\pm.000}$ | $.175_{\pm.001}$ | $.215_{\pm.000}$ |
| | 192 | $.221_{\pm.000}$ | $.255_{\pm.000}$ | $.221_{\pm.000}$ | $.255_{\pm.000}$ |
| | 336 | $.277_{\pm.000}$ | $.295_{\pm.001}$ | $.277_{\pm.001}$ | $.296_{\pm.001}$ |
| | 720 | $.353_{\pm.001}$ | $.348_{\pm.001}$ | $.355_{\pm.000}$ | $.348_{\pm.000}$ |
| | Avg. | $.256_{\pm.000}$ | $.277_{\pm.000}$ | $.257_{\pm.000}$ | $.278_{\pm.000}$ |
| Solar | 96 | $.194_{\pm.005}$ | $.229_{\pm.004}$ | $.207_{\pm.000}$ | $.246_{\pm.001}$ |
| | 192 | $.228_{\pm.002}$ | $.256_{\pm.003}$ | $.239_{\pm.001}$ | $.270_{\pm.001}$ |
| | 336 | $.247_{\pm.006}$ | $.276_{\pm.005}$ | $.260_{\pm.001}$ | $.287_{\pm.001}$ |
| | 720 | $.251_{\pm.003}$ | $.275_{\pm.003}$ | $.264_{\pm.001}$ | $.291_{\pm.001}$ |
| | Avg. | $.230_{\pm.002}$ | $.259_{\pm.002}$ | $.242_{\pm.000}$ | $.274_{\pm.000}$ |
| ECL | 96 | $.139_{\pm.001}$ | $.235_{\pm.002}$ | $.139_{\pm.001}$ | $.233_{\pm.001}$ |
| | 192 | $.160_{\pm.002}$ | $.254_{\pm.002}$ | $.158_{\pm.001}$ | $.249_{\pm.001}$ |
| | 336 | $.176_{\pm.003}$ | $.271_{\pm.003}$ | $.177_{\pm.001}$ | $.271_{\pm.001}$ |
| | 720 | $.198_{\pm.003}$ | $.292_{\pm.006}$ | $.201_{\pm.003}$ | $.293_{\pm.002}$ |
| | Avg. | $.168_{\pm.001}$ | $.264_{\pm.001}$ | $.169_{\pm.001}$ | $.262_{\pm.001}$ |
| Traffic | 96 | $.378_{\pm.000}$ | $.258_{\pm.000}$ | $.378_{\pm.000}$ | $.259_{\pm.000}$ |
| | 192 | $.393_{\pm.001}$ | $.267_{\pm.001}$ | $.399_{\pm.000}$ | $.270_{\pm.000}$ |
| | 336 | $.399_{\pm.001}$ | $.276_{\pm.002}$ | $.416_{\pm.001}$ | $.279_{\pm.000}$ |
| | 720 | $.437_{\pm.001}$ | $.289_{\pm.002}$ | $.456_{\pm.001}$ | $.297_{\pm.001}$ |
| | Avg. | $.402_{\pm.000}$ | $.273_{\pm.001}$ | $.412_{\pm.000}$ | $.276_{\pm.000}$ |

Table N.3: Results of TS forecasting over five runs - 3) Other datasets.

