# OpenReview forum: "Sequential Order-Robust Mamba for Time Series Forecasting"
_ICLR.cc/2025/Conference — Submitted to ICLR 2025_

### Official Review · Reviewer_Cgt6 · 2024-10-29

**Soundness:** 2
**Presentation:** 3
**Contribution:** 2
**Rating:** 6
**Confidence:** 4

**Summary:**

In this manuscript, the authors claim that channels in time series (TS) data have no specific order in general, recent studies have adopted
Mamba to capture channel dependencies (CD) in TS, introducing a sequential order bias. To address this issue, the authors propose SOR-Mamba, a TS forecasting method that 1) incorporates a regularization strategy to minimize the discrepancy between
two embedding vectors generated from data with reversed channel orders, thereby enhancing robustness to channel order.

**Strengths:**

1. The authors propose SOR-Mamba, a TS forecasting method that handles the sequential order bias by
1) regularizing the unidirectional Mamba to minimize the distance between two embedding vectors
generated from data with reversed channel orders for robustness to channel order.
2. The authors introduce CCM, a novel pretraining task that preserves the correlation between channels from
the data space to the latent space, thereby enhancing the model’s ability to capture CD.
3. With the exception of the power-related datasets (ETTm1 & m2, etc.), the experimental performance exceeds the current SOTA.

**Weaknesses:**

1. It is puzzling why the authors mention 'robust' but does not give experiments to resist noise or PGD attacks. Or it may be necessary to modify the terminology in the manuscript to be more precise, after all, 13 different domains of time series prediction tasks have been tried.
2. With the exception of the power-related datasets (ETTm1 & m2, etc.), the experimental performance exceeds the current SOTA. Why is the performance of the author's scheme inferior to SOTA (i.e., PatchTST) on power-related datasets (ETTm1 & m2, etc.) with more significant global regularity?
3. Despite the computational complexity comparison, the reviewer is concerned about the actual training and inference time.
4. The description in Figure 1 seems to be the difference between different exogenous time series due to sampling frequency and periodic change rule, and the description of sequence order is prone to misunderstanding (i.e., covariables have orders).

**Questions:**

Please see weaknesses.

---

> ### Author Response · Authors · 2024-11-15
>
> ###  **W1.  Definition of robustness**
>
>
> We understand that robustness refers to the **stability of results or performance despite variations in certain factors**, and is not solely related to the context of adversarial attacks, as the reviewer mentioned. As noted in the **L17**, the robustness we discuss specifically pertains to **'robustness to channel order'** in TS, particularly regarding the channel order that Mamba receives as input
>
>
>
>
> &nbsp;
>
> ----
> ### **W2.  Performance of ETT datasets**
>
>
> It is worth noting that
> (1) CD models, which capture dependencies between channels, typically have high capacity but lower robustness, whereas CI models exhibit the opposite [A,B]; and
> (2) the ETT datasets have the relatively small size among the benchmark datasets, as shown in **Appendix A.1**.
>
>
> Due to these factors, CI models (e.g., PatchTST [C], RLinear [D]) generally outperform CD models (e.g., iTransformer [E], S-Mamba [F], SOR-Mamba [ours]) on smaller datasets, including the ETT datasets. We believe that the relatively lower performance on the ETT datasets is not due to regularization strategy, but rather to the smaller dataset size, where CI models tend to be more suitable.
>
> &nbsp;
>
> [A] “The Capacity and Robustness Trade-off: Revisiting the Channel Independence Strategy for Multivariate Time Series.” TKDE (2024)
>
> [B] Nie, Tong, et al. "Channel-aware low-rank adaptation in time series forecasting." CIKM (2024)
>
>
> [C] Nie, Yuqi, et al. "A time series is worth 64 words: Long-term forecasting with transformers." ICLR (2023)
>
>
> [D] Li, Zhe, et al. "Revisiting long-term time series forecasting: An investigation on linear mapping." arXiv (2023)
>
>
> [E] Liu, Yong, et al. "iTransformer: Inverted transformers are effective for time series forecasting." ICLR (2024)
>
>
> [F] Wang, Zihan, et al. "Is mamba effective for time series forecasting?." arXiv (2024)
>
>
> &nbsp;
>
> ----
> ### **W3.  Training and inference time**
> We appreciate the reviewer's interest in the practical efficiency of our approach. However, we have ***already*** conducted an analysis regarding the efficiency in terms of (1) the number of parameters, (2) complexity and memory, and (3) computational time (for both training and inference) using the Traffic dataset, which has a large number of channels (C=862), as shown in **Table 13**.
>
>
>
> &nbsp;
>
> ----
> ### **W4.  Misunderstanding of the term “sequence order”**
> Thank you for pointing this out. The varying sampling frequencies and periodic change rules for different channels in **Figure 1** were intended solely to illustrate our motivation in representing distinct channels. We agree that this may be confusing, so we will add further explanations to clarify this point.

---

> ### Comment · Reviewer_Cgt6 · 2024-11-15
> **Feedback for authors**
>
> Dear Authors:
>
> Thank you for your rebuttal!
>
> A1: The author also claims that this is the property of stability of model after some factors are affected, so according to the description of the property of the neural network in the statistical learning book, this should be called stability. This is mainly because robustness is too easily understood to mean that the model performance does not degrade after being disturbed by input noise.
>
> A2: If the data set size is small (i.e., ETT dataset), then the proposed method seems to have under-fitting on small-scale tasks, which will be an obstacle for the model to solve small-scale time series forecasting tasks.
>
> I was generally satisfied with the rest of the responses. Thus, I'm willing to increase the score to 6.

---

> > ### Author Response · Authors · 2024-11-15
> >
> > Thank you for raising your score!
> >
> > If you have any further questions or suggestions, please feel free to share them with us.

---

### Official Review · Reviewer_vPYA · 2024-10-31

**Soundness:** 2
**Presentation:** 3
**Contribution:** 2
**Rating:** 5
**Confidence:** 4

**Summary:**

This paper introduce a new Mamba-based model for long term time series forecasting named SOR-Mamba which utilizes modified Mamba structure to capture channel dependencies. Furthermore, the authors also introduces channel correlation modeling as a pretraining task aimed at enhancing the ability to capture channel dependencies.

**Strengths:**

1. The proposed SOR-Mamba uses a modified Mamba structure (also with a regularization strategy) to eliminate the sequential order bias of Mamba which is more useful in capture channel dependencies.
2. The authors also introduce a pretraining task named channel correlation modeling (CCM) to preserve correlations between channels from the data space to the latent space in order to enhance the ability to capture CD.

**Weaknesses:**

1. Since Mamba is a sequential-based model, and the channels lake sequential order, why we need Mamba to extract channel dependencies? It is important to clarify the importance of capturing channel dependencies with Mamba.
2. The experimental comparisons appear unfair, as models like PatchTST exhibit better performance with longer lookback window. I strongly suggest the authors conduct additional experiments, such as setting $L=336$.

**Questions:**

See Weaknesses above.
3. In line 369-370, I’m curious about how to calculate the global correlation.

**Details Of Ethics Concerns:**

I think these if no ethics concerns.

---

> ### Author Response · Authors · 2024-11-15
>
> ### **W1. Why Mamba for capturing CD?**
>
>
> As noted in **Line 39–41**, ***our focus is on Mamba’s role in capturing CD, in line with recent work [A] that advocates for using complex attention mechanisms for CD while employing simple multi-layer perceptrons (MLPs) for TD.*** Consequently, recent works [B,C] have also explored using Mamba as an alternative to attention for capturing CD, aligning with our direction. In addition, we present further experiments on using Mamba to capture temporal dependency (TD) in **Table 11**. These results are consistent with previous studies, suggesting that complex architectures like attention or Mamba are more effective for capturing CD than TD.
>
> &nbsp;
>
> [A] Liu, Yong, et al. "iTransformer: Inverted transformers are effective for time series forecasting." ICLR (2024)
>
> [B] Wang, Zihan, et al. "Is mamba effective for time series forecasting?." arXiv (2024)
>
> [C] Ma, Shusen, et al. "Fmamba: Mamba based on fast-attention for multivariate time-series forecasting." arXiv (2024)
>
> &nbsp;
>
>
> ---
> ###  **W2.  Longer lookback window ($L$)**
>
> Thank you for your insightful comment.
>
> We set $L=96$ to align with the experimental protocols used in previous works (iTransformer [A], S-Mamba [B]). However, as the reviewer suggested, we conducted an **additional ablation study** on input length across four datasets (Traffic, Electricity, PEMS04, ETTm1), testing various lookback lengths $L \in \\{48, 96, 192, 336, 720\\}$, with forecast horizon of $H= 12$ for PEMS04 and $H = 96$ for the other datasets, in accordance with previous works [A,B]. The results, shown in **Appendix L**, indicate that performance remains robust to the choice of $L$ for some datasets and even improves with a larger $L$ for others.
>
> &nbsp;
>
> [A] Liu, Yong, et al. "iTransformer: Inverted transformers are effective for time series forecasting." ICLR (2024)
>
> [B] Wang, Zihan, et al. "Is mamba effective for time series forecasting?." arXiv (2024)
>
>
> &nbsp;
>
>
> ---
> ###  **W3.  Calculation of (global) correlation**
>
> Thank you for pointing this out.
>
> We agree that the description of calculating the (global) correlation could be clearer. The global correlation, as mentioned in **L369--370**, refers to the correlation between channels across the entire "training" TS dataset, excluding the validation and test datasets. Details on the train-validation-test split protocol are described in **Appendix A**.

---

> ### Comment · Reviewer_vPYA · 2024-11-26
>
> Thanks for the author's responses to my questions. I still think the motivation of using Mamba to capture cross-channel dependencies is unconvincing. Additionally, it would be beneficial to evaluate the impact of the lookback window on other datasets, such as ETTH1. I will keep my original score and discuss with other reviewers.

---

> > ### Author Response · Authors · 2024-11-27
> >
> > Thank you for engaging in the discussion.
> >
> > Below we further clarify the motivation and the additional experimental results.
> >
> > &nbsp;
> >
> > ---
> > ## 1. Motivation of Mamba for CD.
> >
> > tl;dr: While Reviewer vPYA has expressed doubt, ***"using Mamba to capture CDs"*** has been a widely accepted concept in prior/concurrent works [C,D,E,F].
> >
> > We believe ***"using Mamba to capture CDs"*** is a natural extension, based on two key points: 1) Mamba [A] has emerged as a promising alternative to Transformer, and 2) capturing CDs has been shown to be essential in prior works [B,C]; more importantly, ***prior/concurrent works already have widely explored using Mamba to capture CDs [C,D,E,F]***, yielding ***competitive results compared to Transformers***. However, given that Mamba is a sequence-based model, appropriate adjustments are required, where we pose our contribution.
> >
> > Note that we also confirm if Mamba is useful for capturing time dependencies (as a replacement of "MLP for TD" in **Figure 2**) in **Table 11**. This experiment demonstrates that simple MLPs are more effective for TD, consistent with the observation in prior works in this area.
> >
> > &nbsp;
> >
> > [A] Gu, Albert, and Tri Dao. "Mamba: Linear-time sequence modeling with selective state spaces." arXiv (2024)
> >
> > [B] Liu et al. "iTransformer: Inverted transformers are effective for time series forecasting." ICLR (2024)
> >
> > [C] Wang et al. "Is mamba effective for time series forecasting?." arXiv (2024)
> >
> > [D] Liang et al. "Bi-Mamba4TS: Bidirectional Mamba for Time Series Forecasting." arXiv (2024)
> >
> > [E] Weng et al. "Simplified Mamba with Disentangled Dependency Encoding for Long-Term Time Series Forecasting." arXiv (2024)
> >
> > [F] Behrouz, A., Santacatterina, M., & Zabih, R. “Mambamixer: Efficient selective state space models with dual token and channel selection" arXiv (2024)
> >
> > &nbsp;
> >
> > ---
> > ## 2. Additional Experiments
> >
> > In response to the reviewer's previous request, we have conducted experiments in **Appendix L**.
> > We chose **four datasets with varying numbers of channels** ( ETTm1 (7), PEMS04 (180), ECL (321), and Traffic (862)) based on prior works [A,B] (i.e., not arbitrarily chosen). Regarding the newly mentioned ETTh1 dataset, *we initially considered it redundant, as it belongs to the same ETT series as ETTm1*, which we had already tested.
> >
> > Nonetheless, to address the additional request, we performed ***additional experiments on ETTh1*** as well, and the results are presented below. Note that we did not perform the pretraining task (CCM) for SOR-Mamba in order to ensure a fair comparison, as it could potentially lead to further performance gains. Additionally, we used $L=H=96$, as in previous works [A,B] and **Appendix L**, and performed a search over $D \in \\{128, 256, 512\\}$ to find the optimal configuration for both methods. The results below demonstrate **superior performance compared to S-Mamba across all values of $L$**.
> >
> > | L | 48 | 96 | 192 | 336 | 720 |
> > |----------|----------|----------|----------|----------|----------|
> > | a) S-Mamba | .391 | .385 | .390 | .397 | .397 |
> > | b) SOR-Mamba | **.379** | **.379** | **.381** | **.389** | **.386** |
> > | a-b) MSE Imp. | .012  | .006 | .009 | .008 | .011 |
> >
> > [A] Liu, Yong, et al. "iTransformer: Inverted transformers are effective for time series forecasting." ICLR (2024)
> >
> > [B] Wang, Zihan, et al. "Is mamba effective for time series forecasting?." arXiv (2024)
> >
> >
> > &nbsp;
> >
> > We hope this addresses your concerns on the motivation and the additional experiment.
> >
> > If you have any further questions, please feel free to ask; I would appreciate it.

---

> > > ### Comment · Reviewer_vPYA · 2024-12-02
> > >
> > > Thanks for the author's response. I do not think "using mamba to capture CDs" is a natural extension, because mamba is a sequence-based model, thus i believe it is not necessary to capture cross-variable dependencies. In addition, i find that the performance of SOR-Mamba is worse than the results of PatchTST.

---

> > > > ### Author Response · Authors · 2024-12-03
> > > >
> > > > > ### **1. Mamba for CD**
> > > >
> > > > Since Mamba is a **sequential** model, it is understandable that capturing ***temporal dependency (TD) seems natural***, while capturing **channel dependency (CD) might seem awkward** at glance.
> > > >
> > > >
> > > > However, there are two key aspects of Mamba to consider:
> > > > - (1) Mamba is a **sequential model**.
> > > > - (2) Mamba is a powerful and more efficient algorithm that **replaces the attention mechanism** in Transformer models.
> > > >
> > > > While focusing *solely on (1)* may make its use for CD appear unconventional, *emphasizing (2)*—its role as an alternative to attention—clarifies how it captures CD, akin to attention mechanisms. (Note that recently, CD is widely captured using the attention mechanism [A,B,C,D,E])
> > > >
> > > >
> > > > To further motivate Mamba for CD from history, while the attention mechanism in Transformers is inherently not sequential, positional encoding is introduced to ensure sequentiality. Using Mamba to capture CD follows a comparable rationale. Mamba has recently proven to be a promising alternative to Mamba, and it is worth trying Mamba for various applications.
> > > >
> > > > &nbsp;
> > > >
> > > > Moreover, *“the use of Mamba to capture CD”* is **NOT* a novel claim introduced by our work; rather, it is an emerging trend in recent studies [F,G,H,I], yielding competitive results compared to Transformers, and our focus is on ***addressing the challenges within this framework***. Thus, **questioning the fundamental validity of using Mamba for CD lies beyond the scope of our research**.
> > > >
> > > > &nbsp;
> > > >
> > > > [A] Liu, Yong, et al. "iTransformer: Inverted transformers are effective for time series forecasting." ICLR (2024)
> > > >
> > > > [B] Gao, Shanghua, et al. "Units: Building a unified time series model." NeurIPS (2024)
> > > >
> > > > [C] Ilbert, Romain, et al. “SAMformer: Unlocking the Potential of Transformers in Time Series Forecasting with Sharpness-Aware Minimization and Channel-Wise Attention.” NeurIPS (2024)
> > > >
> > > > [D] Wang, Xue, et al. “CARD: Channel aligned robust blend transformer for time series forecasting.” ICLR (2024)
> > > >
> > > > [E] Yu, Guoqi, et al. “Revitalizing Multivariate Time Series Forecasting: Learnable Decomposition with Inter-Series Dependencies and Intra-Series Variations Modeling.” ICML (2024)
> > > >
> > > > [F] Wang et al. "Is mamba effective for time series forecasting?." arXiv (2024)
> > > >
> > > > [G] Liang et al. "Bi-Mamba4TS: Bidirectional Mamba for Time Series Forecasting." arXiv (2024)
> > > >
> > > > [H] Weng et al. "Simplified Mamba with Disentangled Dependency Encoding for Long-Term Time Series Forecasting." arXiv (2024)
> > > >
> > > > [I] Behrouz, A., et al. “Mambamixer: Efficient selective state space models with dual token and channel selection" arXiv (2024)
> > > >
> > > > &nbsp;

---

> ### Author Response · Authors · 2024-12-03
>
> > ### **2. Comparison with PatchTST**
>
> We believe your statement, *"(I find that) the performance of SOR-Mamba is worse than the results of PatchTST"*, is inaccurate, as shown in **Table 2**, where our method outperforms PatchTST on 11 out of 13 datasets.
>
> It is important to note that our work uses the same $L$ for all baseline methods for **fair comparison**, along with previous works. To clarify, we present five representative SOTA algorithms, listed chronologically by their release dates, along with the number of citations (as of 2024.12.02) and their input length ($L$):
> - **[A] PatchTST (ICLR 2023, 1031 citations)**: $L = 512$ (for self-supervised setting) and $L = 336$ (for supervised)
> - **[B] Crossformer (ICLR 2023, 523 citations)**: $L \in \\{24, 48, 96, 168, 336, 720\\}$ (tuned as a hyperparameter)
> - **[C] RLinear (arXiv 2023, 74 citations)**: $L = 336$
> - **[D] iTransformer (ICLR 2024, 432 citations)**: $L = 96$
> - **[E] S-Mamba (arXiv 2024, 19 citations)**: $L = 96$
> - **[Ours] SOR-Mamba**: $L = 96$
>
> These works use different $L$ values, and **aligning them to a unified setting (along with an ablation study of varying $L$) is a common practice**.  Furthermore, it is important to note that our setting follows the experimental settings and analyses established in **recent SOTA work [D,E]**, which is widely used as a backbone for recent methods [F, G], and the comparison with PatchTST is also done within this context.
>
> We respectfully believe that claiming our method underperforms PatchTST overlooks the experimental settings and analyses established in recent SOTA work [D], which are commonly used in contemporary methods [F, G].
>
> &nbsp;
>
> [A] Nie, Yuqi, et al. "A time series is worth 64 words: Long-term forecasting with transformers." ICLR (2023)
>
> [B] Zhang, Yunhao, and Junchi Yan. "Crossformer: Transformer utilizing cross-dimension dependency for multivariate time series forecasting." ICLR (2023)
>
> [C] Li, Zhe, et al. "Revisiting long-term time series forecasting: An investigation on linear mapping." arXiv (2023).
>
> [D] Liu, Yong, et al. "iTransformer: Inverted transformers are effective for time series forecasting." ICLR (2024)
>
> [E] Wang et al. "Is mamba effective for time series forecasting?." arXiv (2024)
>
> [F] Dong, et al. "TimeSiam: A Pre-Training Framework for Siamese Time-Series Modeling." ICML (2024)
>
> [G]  Gao, Shanghua, et al. "Units: Building a unified time series model." NeurIPS (2024)
>
> &nbsp;
>
> ---
> > ### **3. Performance of PatchTST on small-scale datasets**
>
> It seems that the reviewer is particularly interested in the **ETT datasets** (as the reviewer requested an additional ablation study with ETTh1, and PatchTST (CI model) is one of the SOTA models for the ETT datasets). However, ***for other datasets, CD models (including Ours) generally perform better***, demonstrating the effectiveness of these algorithms.
>
> To be precise, according to **Table 10 of iTransformer [A]** and **Table 3 of S-Mamba [B]**, both results show that, for the ETT series (ETTh1, h2, m1, m2), iTransformer and S-Mamba, which are **CD models**, perform worse than both RLinear [C] and PatchTST [D], which are **CI models**.
>
>
> However, it is worth noting that
> - (1) **CD models** typically have high capacity but lower robustness, whereas **CI models** exhibit the opposite [E, F]; and
> - (2) ETT datasets are relatively small compared to other benchmark datasets, as shown in **Appendix A.1**.
>
> We note that this issue regarding the ETT datasets, has already been addressed, which was also raised by Reviewer Cgt6.
>
> &nbsp;
>
>
> [A] Liu, Yong, et al. "iTransformer: Inverted transformers are effective for time series forecasting." ICLR (2024)
>
> [B] Wang et al. "Is mamba effective for time series forecasting?." arXiv (2024)
>
> [C] Nie, Yuqi, et al. "A time series is worth 64 words: Long-term forecasting with transformers." ICLR (2023)
>
> [D] Li, Zhe, et al. "Revisiting long-term time series forecasting: An investigation on linear mapping." arXiv (2023).
>
> [E] Han, L., Ye, H. J., & Zhan, D. C. “The Capacity and Robustness Trade-off: Revisiting the Channel Independence Strategy for Multivariate Time Series.” TKDE (2024)
>
> [F] Nie, Tong, et al. "Channel-aware low-rank adaptation in time series forecasting." CIKM (2024)
>
> &nbsp;
>
> We hope the three responses above address your concerns and provide clarification.

---

> > ### Comment · Reviewer_vPYA · 2024-12-03
> >
> > Thanks for your response. In fact, CD models perform worse than CI models on ETT datasets (maybe weather has the same results), because CD models capture spurious cross-variable feature which i think is a more important question to be solved. In addition, I think the only problem of this work is its motivation. Simply citing previous work (which is even just follow Mamba) cannot convince me. Therefore, the author needs to reorganize the motivation and importance of this work. And I guarantee that I am familiar with this field. I hope the author do not try to just show the advantages of this work (such as better performance on ecl and traffic, but unstable performance on ETTs). I think the experimental results are not important. What is important is the problem that this work is going to solve. Finally, my purpose is to make this work better, not to reject it. Therefore, I encourage the author to think carefully about the motivation of this work, and i am glad to discuss with the author.

---

> > > ### Author Response · Authors · 2024-12-03
> > >
> > > Thank you for engaging in the discussion again!
> > >
> > > &nbsp;
> > >
> > > We first carefully reviewed your questions; as we understand, your concern has shifted from the motivation behind the general idea of **"Mamba for CD"** to the motivation behind **“our work”**.
> > >
> > > > 1st question: *Since Mamba is a sequential-based model, and the channels lake sequential order, why we need Mamba to extract channel dependencies?*
> > >
> > > > 2nd question: *the motivation of using Mamba to capture cross-channel dependencies is unconvincing.*
> > >
> > > > 3rd question: *I do not think "using mamba to capture CDs" is a natural extension, because mamba is a sequence-based model,*
> > >
> > > > 4th question (now): ***the only problem of this work is its motivation. .. What is important is the problem that this work is going to solve.***
> > >
> > > &nbsp;
> > >
> > > We believe your concern on the motivation behind the general idea of **"Mamba for CD"** has been addressed (if not, please let us know asap!), and below we focus on addressing your concern on the motivation behind **“our work”**.
> > >
> > >
> > > To recap the motivation behind **"Mamba for CD"**, Mamba has emerged as a promising alternative to Transformer, and we believe Mamba for CD is a worthy trial, analogous to the variation of Transformer for various types of inputs. For example, Transformers are basically permutation-equivariant and for unordered data (sets), but positional encoding is introduced to make it permutation-aware for ordered data structures, e.g., languages, images, or time series. Analogously, Mamba is basically for ordered (sequential) data, but we are trying to process unordered data (channels).
> > >
> > > As you already know, previous works have attempted to apply Mamba to capture CDs. While they showed competitive performance (often better than Transformer-based methods), we also agree that these attempts might not be fully convincing. Specifically, **our motivation** lies in ***addressing the sequential order bias*** inherent in previous Mamba-based methods.
> > >
> > > &nbsp;
> > >
> > > Below we divide the motivation for **our work** step by step:
> > >
> > > 1. **[Problem definition]** ***Sequential order bias*** (**L42--45**)
> > > - “Applying Mamba to capture CD is challenging as channels lack an inherent sequential order, whereas Mamba is originally designed for sequential inputs (i.e., Mamba contains a sequential order bias)”
> > >
> > > 2. **[Previous works]** Employs ***bidirectional Mamba*** (**L46--51**)
> > > - “To address this issue, previous works have employed bidirectional Mamba to capture CD, where two unidirectional Mambas with different parameters capture CD from a certain channel order and its reversed order”
> > >
> > > 3. **[Limitation of previous works]** (**L51--63**)
> > > - *“These methods are inefficient due to the need for two models”* (**L51--52**).
> > > - *“The table indicates that 1) bidirectional Mamba does not always achieve the best performance, and 2) the performance of unidirectional Mamba varies depending on the channel order”* (**L59--63, Table 1**)
> > >
> > > &nbsp;
> > >
> > > The **“sequential order bias”** issue and the **“limitations of previous works”** in addressing this issue have motivated our work on “Sequential order-robust Mamba,” which utilizes a regularization strategy to replace the usage of bidirectional Mamba. Furthermore, previous works applied 1D-convolution in Mamba, originally designed for sequential data, ***without questioning its suitability for channel data in TS datasets***. This motivated our work to remove it, as channels lack inherent order, which is further discussed in detail in **Appendix D**.
> > >
> > > &nbsp;
> > >
> > > For clarity, we will revise the manuscript to better explain both the motivation for our work and the use of Mamba for CD.
> > >
> > > &nbsp;
> > >
> > > We hope this clarifies the motivation behind our work!

---

### Official Review · Reviewer_9Yaz · 2024-11-01

**Soundness:** 3
**Presentation:** 3
**Contribution:** 3
**Rating:** 6
**Confidence:** 5

**Summary:**

This manuscript proposes an algorithm to regularize bidirectional Mamba channel vectors in order to solve the problem that there is no intrinsic order relationship between channels in time series data. The authors also design a new pre-training method (CCM) to help the model better establish the association between channels of multi-dimensional time series. Extensive experiments and ablation experiments are conducted, and the experimental results show the effectiveness of the relevant contributions.

**Strengths:**

1. This study achieves a good improvement in results with only a few changes to the method. The proposed method is simple, novel and effective.
2. The experiments carried out by the authors are very detailed and the relevant parameters are well discussed. This is an experimentally rigorous and solid study.
3. The manuscript is well-written and the figures are clear and intuitive.

**Weaknesses:**

1. The authors lack a complete description of the motivation for regularization. Why regularizing the bidirectional mamba channel vectors and minimizing the distance between them can help the model better capture the correlation between channels? This requires the author to give a more intuitive and reasonable motivation.
2. Although the authors have done a lot of experiments, they lack experiments on longer input lengths. Some widely used baselines in this field, such as PatchTST (arXiv preprint arXiv:2211.14730, 2022.), all use longer input sequences.
3. The authors use too many abbreviations in their manuscripts and experimental result tables, and lack clear and complete explanations of the abbreviations. This makes the relevant content very difficult to read and easily confusing.

**Questions:**

1. Table I.1 seems to show that the hyperparameter lambda does not affect the results. Is this a reasonable phenomenon for the key contribution?
2. What are the principles and motivations for using parameter-sharing CD-Mamba block and removing 1D-conv? Can the authors provide additional details?
3. The description of the pre-training method used in this study is somewhat vague. Can the authors supplement the algorithm flow or provide detailed explanation?

---

> ### Author Response · Authors · 2024-11-15
>
> ###  **W1.  Motivation for regularization**
>
>
> It is important to clarify that the proposed regularization was ***not specifically intended to enhance the capture of channel correlation***, but to ***address the sequential order bias***.
>
>
> As shown in **Table 1** and **Figure 7**, previous methods employing bidirectional Mamba to handle sequential order bias still exhibited sensitivity to channel order, both in terms of performance and qualitative results.
> This motivated us to develop a more effective approach to address the bias: the regularization strategy. This approach yields a more consistent representation that is robust to changes in channel order, as demonstrated in **Table 10**, **Appendix H.1** and **Figure 7**.
>
>
>
>
> &nbsp;
>
> ---
> ###  **W2.  Longer lookback window**
>
>
> We set $L=96$ to align with the experimental protocols used in previous works (iTransformer [A], S-Mamba [B]). However, as the reviewer suggested, we also conducted an ablation study on input length across four datasets (Traffic, Electricity, PEMS04, ETTm1), testing various lookback lengths $L \in \\{48, 96, 192, 336, 720\\}$, with forecast horizon of $H= 12$ for PEMS04 and $H = 96$ for the other datasets, in accordance with previous works [A,B]. The results, shown in **Appendix L**, indicate that performance remains robust to the choice of $L$ for some datasets and even improves with a larger $L$ for others.
>
>
> &nbsp;
>
>
> [A] Liu, Yong, et al. "iTransformer: Inverted transformers are effective for time series forecasting." ICLR (2024)
>
> [B] Wang, Zihan, et al. "Is mamba effective for time series forecasting?." arXiv (2024)
>
>
>
>
> &nbsp;
>
> ---
> ###  **W3.  Clarity of sentences**
>
>
> Thank you for your feedback.
>
>
> We understand that the excessive use of abbreviations without clear explanations can make the manuscript and experimental result tables difficult to follow.
>
>
> Below is a list of the abbreviations used:
> - CI/CD: channel independence/channel dependence
> - TD: temporal dependencies
> - LTSF: long-term TS forecasting
> - 1D-conv: 1D-convolution
> - SSL/SL: self-supervised learning / supervised learning
> - FT/LP: fine-tuning / linear probing,
> - MSE/MAE: mean squared error / mean absolute error
>
>
> We believe these abbreviations are commonly used in previous works. However, we will replace the abbreviations (TD, FT, LP) with full expressions, as they are not widely recognized, and expand (LTSF), as it appears infrequently in our manuscript. We will revise the text to provide clearer definitions and reduce abbreviation use to improve readability.

---

> ### Author Response · Authors · 2024-11-15
>
> ###  **Q1.  Hyperparameter lambda and the effect of regularization**
>
>
> As the reviewer mentioned **Table I.1** illustrates that the inclusion of the regularization loss significantly improves the performance **regardless of the value of lambda**.
>
>
> This phenomenon is reasonable, as shown in **Figure 9** and **Figure G.1**, where the regularization loss converges very quickly early in the training process for our proposed methods. This rapid convergence suggests that the regularization effectively aligns the two reversed-order vectors with ease, yielding consistent performance improvements regardless of the value of lambda.
>
> &nbsp;
>
>
> ---
> ###  **Q2.  Motivation of (1) single unidirectional Mamba and (2) removal of 1D-conv.**
>
>
> > (1) **Motivation of single unidirectional Mamba**
>
>
> Previous research used two separate Mambas to capture channel dependencies from both directions [A, B] in an effort to address the channel order bias. However, we found that using bidirectional Mamba does not always yield the best performance (**Table 1**) and can be inefficient (**L51--52**). By using a single Mamba with regularization loss, we effectively address this bias, achieving superior performance (**Table 2,4,5**)  and improved efficiency (**Table 13**) compared to using two Mambas."
>
> &nbsp;
>
> > **(2) Motivation of removal of 1D-conv**
>
>
> The 1D-conv in Mamba [C] was originally designed to capture local information in sequential data (**L66--68**), making it less suitable for non-sequential data (i.e., channels in TS). However, previous works have overlooked the need to remove the 1D-conv when using Mamba to capture channel dependencies (CD) [D,E,F], where channels do not have sequential order. This has motivated us to remove it from the Mamba block. Further details on the removal of the 1D-conv are provided in **Appendix D**.
>
>
> &nbsp;
>
>
> [A] Wang, Zihan, et al. "Is mamba effective for time series forecasting?." arXiv (2024)
>
>
> [B] Liang, Aobo, et al. "Bi-Mamba4TS: Bidirectional Mamba for Time Series Forecasting." arXiv (2024)
>
>
> [C] Gu, Albert, and Tri Dao. "Mamba: Linear-time sequence modeling with selective state spaces." arXiv (2024)
>
>
> [D] Weng, Zixuan, et al. "Simplified Mamba with Disentangled Dependency Encoding for Long-Term Time Series Forecasting." arXiv (2024)
>
>
> [E] Cai, Xiuding, et al. "MambaTS: Improved Selective State Space Models for Long-term Time Series Forecasting." arXiv (2024)
>
>
> [F] Ma, Shusen, et al. "Fmamba: Mamba based on fast-attention for multivariate time-series forecasting." arXiv (2024)
>
> &nbsp;
>
>
> ---
> ###  **Q3. Lack of details regarding CCM**
>
>
> We apologize for the lack of detail regarding the proposed pretraining task, CCM. To address this, we have included pseudocode for the task in **Appendix J** and provided additional guidance for readers in **L244**.

---

### Official Review · Reviewer_eFUR · 2024-11-03

**Soundness:** 3
**Presentation:** 3
**Contribution:** 3
**Rating:** 6
**Confidence:** 3

**Summary:**

This paper addresses a limitation in applying Mamba to time series forecasting: the sequential order bias when modelling channel dependencies, as channels typically lack inherent order. The authors propose SOR-Mamba, which introduces two main innovations: a regularization strategy that minimizes discrepancy between embedding vectors from reversed channel orders, and the removal of the unnecessary 1D-convolution layer. They also introduce Channel Correlation Modelling (CCM), a pretraining task that preserves channel correlations from data space to latent space. The method reported promising performance across different scenarios, including cases with missing data and varying channel orders, while maintaining computational efficiency in terms of memory and runtime.

**Strengths:**

The proposed method addresses the sequential order bias in channel dependencies (CD). This bias can degrade performance when channels lack inherent order in multi-channel time series data. The SOR-Mamba method mitigates this bias by introducing a regularization technique that minimizes the discrepancy between embeddings generated from reversed channel orders, thus improving robustness.

Additionally, the paper proposes Channel Correlation Modeling (CCM) as a pretraining task, which preserves channel correlations from data to latent space, enhancing CD capture. The architecture modifications, including removing the 1D-convolution and focusing on unidirectional Mamba, can reduce computational overhead, achieving efficiency gains in memory and parameter usage compared to other models like S-Mamba. This should be of interests to the group of readers working on multi channel time series data forecasting.

**Weaknesses:**

The paper’s approach is relatively complex, involving several architectural changes (e.g., removing 1D-convolutions, applying specific regularization, and adding CCM pretraining) that may complicate replication and limit accessibility.

While it emphasizes efficiency improvements, the paper could provide more detailed explanations regarding the trade-offs involved in removing the 1D-convolution, especially on datasets where channel order may have some inherent structure (e.g., traffic data). The study also lacks a comprehensive discussion of potential limitations in real-world deployment, such as sensitivity to hyperparameters (e.g., λ in regularization) and its impact across different datasets.

**Questions:**

1. How does the choice of the distance metric d in the regularization term impact the model's performance across datasets with varying levels of channel correlation? - is the regularization term’s effectiveness dependent on the number of channels or dataset-specific factors etc.?

2. The paper uses CCM to preserve channel correlations in the latent space during pretraining. How does CCM compare to masked modelling and reconstruction pretraining across different dataset sizes and channel configurations? - and is there any limitation of CCM if there's only exist weak channel correlations?

---

> ### Author Response · Authors · 2024-11-15
>
> ###  **W1.  Complexity of proposed architecture**
> Thank you for your feedback regarding the complexity of our approach. We believe the proposed method is both simple and effective, as noted by reviewer 9Yaz.
>
> &nbsp;
>
> > **(1) Simple architecture**
>
> The removal of the 1D-conv is straightforward and does not require any additional changes to the remaining architecture, as detailed in **Appendix D**.
> For the proposed regularization, we utilize embeddings from the existing architecture without introducing any additional modules or embeddings, as shown in **Appendix C**. This approach preserves simplicity and effectiveness while minimizing computational overhead.
>
> &nbsp;
>
> > **(2) Simple pretraining task**
>
> For the CCM pretraining task, only the correlation between output vectors needs to be computed, with no need for extra modules. To clarify the process, we have included pseudocode for CCM in **Appendix J** and further guidance in **L244**, highlighting the task’s simplicity.
>
> &nbsp;
>
> To facilitate replication and ensure accessibility, we have also provided the code in the supplementary material.
>
>
> &nbsp;
>
> ---
> ###  **W2.  Trade-off in removing 1D-conv**
>
>
> It is important to clarify that ***our primary motivation for removing the 1D-conv is not efficiency***, but rather to address ***general TS datasets***, where channels typically do not have a sequential order. We believe it is reasonable to design a model architecture suited to these general cases.
>
>
> As shown in **Table 7** and **Figure 5**, removing the 1D-conv of Mamba in datasets where channel order has inherent structure (e.g., traffic data) does not always lead to performance benefits. However, this finding further supports our approach, indicating that the 1D-conv captures sequential relationships between tokens (channels). When channels have an inherent order, which is identifiable based on the dataset domain or type, removing the 1D convolution may not be desirable.
>
>
> Further details on the rationale and process for removing the 1D-conv are provided in **Appendix D**.
>
>
> &nbsp;
>
>
>
> ---
> ###  **W3.  Hyperparameter sensitivity**
> Thank you for pointing this out. We have, however, ***already*** discussed the robustness of our proposed method to hyperparameters across various datasets, as follows:
> - Robustness to **hyperparameter $\lambda$**: **Appendix I.1**
> - Robustness to **distance metric (for regularization)**: **Appendix K.1** (as guided in **L207**)
> - Robustness to **distance metric (for CCM)**: **Appendix K.2** (as guided in **L224--255**)
>
>
> &nbsp;
>
> ---
> ###  **Q1.  Robustness to distance metric (for regularization)**
>
>
>
>
>  **Q1-a.  Relationship between the distance metric $d$ and the performance**
>
>
>
> As discussed in **Appendix K.1**, the performance gains from regularization loss are **robust across different choices of the distance metric $d$**. However, we did not observe a consistent relationship between the choice of metric and the specific characteristics of the TS datasets.
>
> &nbsp;
>
>
> **Q1-b.  Relationship between the effectiveness of regularization and the characteristic of dataset**
>
>
> While **Figure 4** shows that the degree of bias varies with the number of channels and channel correlation, we believe that the relationship between performance gains (i.e., the effectiveness of the regularization) and bias reduction is influenced by various inherent factors.
>
>
> &nbsp;
>
>
> ---
> ###  **Q2.  Effect of pretraining tasks by dataset characteristics**
>
> We agree on the importance of analyzing the performance gains of different pretraining tasks and dataset characteristics, particularly in terms of the number of channels. For this reason, we have ***already*** discussed this in **Figure 6**, which demonstrates that the proposed pretraining tasks outperform both MM and reconstruction tasks across datasets with both large and small numbers of channels. Additionally, as shown in the right panel of **Figure 4**, the correlation between channels varies across datasets, with the ETT datasets exhibiting low correlation. Despite this, our method performs well on these datasets, demonstrating its robustness and effectiveness even in cases with weak channel correlations.

---

> > ### Comment · Reviewer_eFUR · 2024-11-25
> >
> > Thanks for the author's responses to my questions and concerns, it seems some of my concerns can be addressed by the results in the appendix, which I didn't read before. Since I already gave a positive rating, I will keep it.

---

### Official Review · Reviewer_WEwK · 2024-11-03

**Soundness:** 3
**Presentation:** 3
**Contribution:** 2
**Rating:** 5
**Confidence:** 2

**Summary:**

This paper introduces SOR-Mamba, a time series forecasting method that addresses the sequential order bias  while capture channel dependencies in Mamba, through a regularization strategy to minimize the discrepancy between two embedding vectors generated from data with reversed channel orders, and 1D-convolution removal, The authors also propose Channel Correlation Modeling (CCM) as a pretraining task.

**Strengths:**

1. Clear problem identification regarding Mamba's limitations in handling unordered channels
2. Comprehensive experiments
3. Improved efficiency compared to other approaches

**Weaknesses:**

**Limited Technical Novelty**:
- The regularization strategy is overly simplistic that minimizing the distance between embeddings from different channel orders
- The removal of 1D-conv lacks theoretical justification and appears to be an ad-hoc solution, As shown in Figure 5 of the paper, its removal may negatively impact datasets with ordered channels such as PEMS datasets

**Questions:**

1. In the Manba architecture, are there other advances in order-invariant architectures that can be compared?
2. The definition of sequential order bias in Figure 4 and how it is calculated?

---

> ### Author Response · Authors · 2024-11-15
>
> ###  **W1.  The regularization strategy is overly simplistic**
>
> We believe that the regularization strategy ***should be simple and efficient*** to effectively replace the bidirectional Mamba for capturing CD. Our proposed regularization approach is intentionally straightforward and efficient, leveraging embeddings directly from the existing architecture ***without requiring additional modules or embeddings***. This design minimizes computational overhead while providing effective regularization within the model.
>
> &nbsp;
>
> ---
> ###  **W2. Justification of removal of 1D-conv**
>
> It is important to note that we remove 1D-conv primarily ***for general TS datasets, where channels do not inherently have a sequential order***. We believe it is reasonable to design a model architecture suited to these general cases.
>
>
> As shown in **Table 7** and **Figure 5**, removing the 1D-conv of Mamba in datasets where channel order has inherent structure (e.g., traffic data) does not always lead to performance benefits. However, this finding further supports our approach, indicating that the 1D-conv captures sequential relationships between tokens (channels). When channels have an inherent order, which is identifiable based on the dataset domain or type, removing the 1D convolution may not be desirable.
>
>
> Further details on the rationale and process for removing the 1D-conv are provided in **Appendix D**.
>
>
> &nbsp;
>
> ---
> ###  **Q1.  Comparison with other (Order-invariant) Mamba architectures**
> As mentioned in **Section 2. Related Works**, several concurrent studies have utilized Mamba to capture CD [A,B,C,D], and these methods generally address the sequential order bias by employing the **bidirectional Mamba [A,B,C,D]**. However, aside from our baseline method [A], most of these works have not released code for reproducibility, making direct comparisons with our approach currently infeasible. We look forward to conducting comparisons when these implementations become available.
>
> &nbsp;
>
>
> [A] Wang, Zihan, et al. "Is mamba effective for time series forecasting?." arXiv (2024)
>
> [B] Liang, Aobo, et al. "Bi-Mamba4TS: Bidirectional Mamba for Time Series Forecasting." arXiv (2024)
>
> [C] Weng, Zixuan, et al. "Simplified Mamba with Disentangled Dependency Encoding for Long-Term Time Series Forecasting." arXiv (2024)
>
> [D] Behrouz, A., Santacatterina, M., & Zabih, R. “Mambamixer: Efficient selective state space models with dual token and channel selection" arXiv (2024)
>
> &nbsp;
>
> ---
> ###  **Q2.  Definition of sequential order bias**
>
> As mentioned in **L322--333**, the sequential order bias is quantified by ***measuring the performance difference (average MSE across four horizons) when the channel order is reversed***, using SOR-Mamba without regularization.

---

### Official Review · Reviewer_PJuw · 2024-11-04

**Soundness:** 3
**Presentation:** 3
**Contribution:** 2
**Rating:** 5
**Confidence:** 4

**Summary:**

This paper proposes SOR-Mamba to solve the sequential order bias introduced by mamba capturing channel dependencies. Model SOR-Mamba incorporates a regularization strategy to minimize the distance between embeddings from data and reversed data and eliminates the 1D-convolution in the original Mamba block. And a pretraining task channel correlation modeling is introduced to preserve the channel correlation from the data space to the latent space. The effectiveness of the proposed method is demonstrated by the extensive experiment results.

**Strengths:**

* This paper uses a single unidirectional Mamba with regularization and non-1D convolutions to capture channel correlations. By modifying the traditional Mamba module, it can better address the issue of sequential order bias. The research problem is clearly defined, and the solution is reasonable. And the pretraining task channel correlation modeling also enhances the model's generalization and performance.

* The details of the model are mostly clear, with each module explained and supported by equations. The experiments  include time series forecasting, transfer learning and many ablation studies to validate the effectiveness of each module.

**Weaknesses:**

* The proposed method is mostly constructed on existing models -- reverse modeling has been widely used in literature and removing the 1d-conv is trival.
* The experimental results show that the model's performance improvement across various datasets is not significant, mostly in the thousandth.
* The removal of 1D-conv negatively impacts PEMS dataset in tale 7 and figure 5. The necessity to remove 1D-conv is uncertain.
* The model uses the Mamba backbone, but the baseline only includes a single Mamba model: S-Mamba. Providing more models from the Mamba framework for comparison would be better.

**Questions:**

* When randomly shuffling the channel order, how is it done? Is a fixed shuffling method used for all sequences in a dataset, or is it shuffled randomly each time?
* In the last row of Table 1, is the calculation formula written incorrectly?

---

> ### Author Response · Authors · 2024-11-15
>
> ###  **W1-a) Reverse modeling is widely used**
>
> We acknowledge that reverse modeling (bidirectional Mamba) is widely used, as noted in **L46--51**. However, we highlight its inherent limitations in **Table 1**, where the bidirectional Mamba does not consistently yield the best performance and, in some cases, underperforms compared to a single unidirectional Mamba. Additionally, the bidirectional Mamba uses two Mambas for tuning, doubling the computational burden compared to a single Mamba. In light of these limitations, we propose an ***alternative to reverse modeling (bidirectional Mamba)***: a single unidirectional Mamba with regularization, which demonstrates improved performance and efficiency, as shown in **Table 2,5** and **Table 13**, respectively.
>
> &nbsp;
>
>
> ---
> ###  **W1-b) Removal of 1D-conv is trivial**
>
> While removing the 1D-conv might seem minor, many studies in the TS domain often adopt architectures directly from NLP or vision tasks **without accounting for TS-specific properties**. Since the 1D-conv in Mamba [A] was originally designed to capture sequential patterns (**L66--68**), applying it to non-sequential TS channels is an **unnatural choice**, yet it has been commonly used without adaptation.  ***We believe our approach is non-trivial, as it specifically accounts for the distinctive characteristics of TS datasets that have been overlooked in other Mamba-based methods. Furthermore, the proposed method has the potential for application in other domains involving non-sequential data (e.g., tabular data)***.
>
> Further details on the removal of the 1D-conv are provided in **Appendix D**.
>
>
> &nbsp;
>
> [A] Gu, Albert, and Tri Dao. "Mamba: Linear-time sequence modeling with selective state spaces." arXiv (2024)
>
> &nbsp;
>
> ---
> ###  **W2.  Performance improvement is not significant.**
> It is important to clarify that our primary goal is not to achieve substantial performance gains over the baseline, but rather to demonstrate that a ***single unidirectional Mamba with a regularization strategy can effectively replace the widely used bidirectional Mamba***, yielding comparable or even better performance with greater efficiency, as shown in **Table 5** and **Table 13**. Additionally, applying the regularization strategy to the bidirectional Mamba also results in performance gains, as shown in **Table 6**.
>
>
> While the optimal model structure (and hyperparameters) may vary between the unidirectional and bidirectional Mamba setups, we used the original hyperparameters from the bidirectional Mamba for the unidirectional Mamba (see **L265--266**), allowing us to evaluate results on the same basis. Optimizing these hyperparameters specifically for the unidirectional Mamba configuration could likely yield additional performance gains.
>
>
> &nbsp;
>
> ---
> ###  **W3.  Necessity of removing 1D-conv**
>
>
> It is important to note that we remove 1D-conv primarily ***for general TS datasets, where channels do not inherently have a sequential order***. We believe it is reasonable to design a model architecture suited to these general cases.
>
>
> As shown in **Table 7** and **Figure 5**, removing the 1D-conv of Mamba in datasets where channel order has inherent structure (e.g., traffic data) does not always lead to performance benefits. However, this finding further supports our approach, indicating that the 1D-conv captures sequential relationships between tokens (channels). When channels have an inherent order, which is identifiable based on the dataset domain or type, removing the 1D convolution may not be desirable.
>
>
> Further details on the rationale and process for removing the 1D-conv are provided in **Appendix D**.
>
>
> &nbsp;
>
> ---
> ### **W4.  Comparison with other Mamba models**.
>
> As mentioned in **Section 2. Related Works**, several concurrent studies have utilized Mamba to capture CD [A,B,C,D], and these methods generally address the sequential order bias by employing the **bidirectional Mamba [A,B,C,D]**. However, aside from our baseline method [A], most of these works have not released code for reproducibility, making direct comparisons with our approach currently infeasible. We look forward to conducting comparisons when these implementations become available.
>
> &nbsp;
>
> [A] Wang, Zihan, et al. "Is mamba effective for time series forecasting?." arXiv (2024)
>
> [B] Liang, Aobo, et al. "Bi-Mamba4TS: Bidirectional Mamba for Time Series Forecasting." arXiv (2024)
>
> [C] Weng, Zixuan, et al. "Simplified Mamba with Disentangled Dependency Encoding for Long-Term Time Series Forecasting." arXiv (2024)
>
> [D] Behrouz, A., Santacatterina, M., & Zabih, R. “Mambamixer: Efficient selective state space models with dual token and channel selection" arXiv (2024)

---

> ### Author Response · Authors · 2024-11-15
>
> ###  **Q1.  Channel order shuffling method (fixed shuffling method vs. shuffled randomly each time)**
>
>
> We use a **fixed shuffling approach** for channel order, with results averaged over five different random seeds. Using a **random shuffling each time** would impair the model’s ability to distinguish between channels, leading to poor performance, as discussed in **Table 12**, **Figure 9**, and **L466–470**.
>
>
> &nbsp;
>
> ---
> ###  **Q2. Incorrect calculation formula (Last row of Table 1)**
>
> Thank you for pointing this out. We initially reversed the sign of the last row of **Table 1** to indicate the degree of **improvement**, as a lower MSE indicates better performance. However, following the reviewer’s feedback, we corrected the sign to avoid any confusion.

---

### Author Response · Authors · 2024-11-23

### **7. Minor issues**
- **Calculation of global correlation (vpYA)**:  The global correlation refers to the correlation between channels across the entire training TS dataset, excluding the validation and test datasets, with details on the train-validation-test split protocol provided in **Appendix A**.

- **Simplicity of the proposed method: (eFUR, WEwK)**: The removal of the 1D-conv is straightforward, requiring no changes to the existing architecture (**Appendix D**), while the regularization uses existing embeddings, maintaining simplicity and minimizing computational overhead (**Appendix C**). Similarly, the CCM pretraining task only computes the correlation between output vectors, with no additional modules (**Appendix J**).

- **Training and inference time (Cgt6)**: We have already analyzed efficiency in terms of (1) parameters, (2) complexity and memory, and (3) computational time (training and inference) using the Traffic dataset ($C=862$), as shown in **Table 13**.

- **Lack of details regarding CCM (eFUR)**: We have included pseudocode for CCM in **Appendix J** and provided additional guidance for readers in **L244**.

&nbsp;

In what follows, we summarize additional changes in the manuscript we provided during the rebuttal period:
- **(Figure L.1; 9Yaz, vPYA)** Applying our method with various lookback windows $L$
- **(Algorithm 2; 9Yaz)** Pseudocode for CCM

&nbsp;

For your convenience, we have uploaded a PDF file that includes both the revised manuscript and appendices, with the major changes highlighted **in green**.

We believe that the discussions and empirical results presented are valuable and will further enhance our contribution. If we have overlooked anything or if you have additional questions or suggestions, please feel free to share them with us; we would be happy to address them.

&nbsp;

Thank you very much.

---

### Author Response · Authors · 2024-11-23

# General Comment

&nbsp;

Dear AC and reviewers,

First of all, we deeply appreciate your time and effort in reviewing our paper. Our work introduces **SOR-Mamba**, a TS forecasting method to address the sequential order bias when capturing channel dependencies (CD) with Mamba. SOR-Mamba utilizes two strategies: **(1) regularization strategy** to minimize the discrepancy between two embedding vectors generated from data with reversed channel orders, and **(2) removal of 1D-convolution** originally designed to capture local information in sequential data.


As highlighted by the reviewers, our work effectively addresses the sequential order bias in Mamba (PJuw, eFUR, Cgt6) using novel regularization and the removal of 1D convolutions (PJuw, WEwK, vPYA). The introduction of Channel Correlation Modeling (CCM) improves the ability to capture channel dependencies (WEwK, eFUR, vPYA, Cgt6). Extensive experiments demonstrate the model's effectiveness and efficiency (PJuw, WEwK, eFUR, 9Yaz).


&nbsp;

In our responses, we addressed the concerns raised by all reviewers and supplemented our claims with additional analyses. Here we provide highlights to aid your post-rebuttal discussion (all sections/figures/tables labeled based on the revision):

&nbsp;

### **1. Motivation of Mamba for CD (vPYA)**

As noted in **L39–41**, our focus is on Mamba’s role in capturing CD, consistent with recent works advocating ***complex attention mechanisms for channel dependency (CD)*** and *simple MLPs for temporal dependency (TD)*. Similarly, **Table 11** confirms that complex architectures like attention or Mamba are better suited for CD than TD.

&nbsp;

### **2. Motivation of the proposed method**

 The reviewers expressed concerns about the motivation and necessity of the proposed methods.

- **Regularization method (9Yaz)**: The regularization was designed to address the sequential order bias, as previous methods using bidirectional Mamba still showed sensitivity to channel order (**Table 1**, **Figure 7**). Our approach provides a more consistent representation, as shown in **Table 10**, **Appendix H.1**, and **Figure 7**.

- **Removal of 1D-conv (9Yaz, PJuw, WEwK)**: The 1D-conv in Mamba was intended for sequential data (**L66--68**) but is not suitable for non-sequential channel data in TS. Previous works overlooked this, which motivated us to remove it, with details in **Appendix D**.

&nbsp;

### **3. Effect of performance by removal of 1D-convolution (PJuw, eFUR, WEwK)**

The 1D-conv is removed for **general** TS datasets where channels lack inherent sequential order. As shown in **Table 7** and **Figure 5**, removing it does not always improve performance in datasets with ordered channels, highlighting that the 1D-conv captures sequential relationships. Removing it may not be beneficial when channel order exists, as discussed in **L362--366**.

&nbsp;

### **4. Comparison with other Mamba-based models (PJuw,WEwK)**
As noted in **Section 2**, several studies have used Mamba to capture CDs. However, these works are concurrent to ours (e.g, TimeMachine (ECAI-2024), S-Mamba, CMamba, FMamba, SST, Bi-Mamba+, MambaTS (Arxiv 2024)) and, aside from our baseline, most have not released their code, making direct comparisons infeasible.

&nbsp;

### **5. Various lookback windows $L$ (9Yaz, vPYA)**
We followed the experimental protocols of previous works ($L=96$) for fair comparison, instead of utilizing different values of $L$. Nonetheless, we conducted an additional ablation study on $L$ across four datasets in **Appendix L** during the rebuttal. The results demonstrate that performance is robust to $L$ for some datasets and improves with a larger $L$ for others.

&nbsp;

### **6. Robustness to hyperparameters (eFUR)**

We have ***already*** discussed the robustness of our proposed method to hyperparameters across various datasets, as follows:

- Hyperparameter λ: **Appendix I.1**

- Distance metric (for regularization): **Appendix K.1**

- Distance metric (for CCM): **Appendix K.2**

---

### Meta-Review · Area_Chair_n14q · 2024-12-20

**Metareview:**

This paper studies an issue when using the Mamba (instead of Transformer) model for time series forecasting. In particular, a sequential order bias was introduced in recent studies when using Mamba to capture channel dependencies in time series. A Mamba variant is proposed in this paper to make the model less dependent on the channel order by incorporating two techniques, namely, regularization strategy and removal of 1D-convolution.

Major strengths:
- The two techniques proposed to make Mamba less dependent on the channel order makes sense although they are quite simple.
- The paper is generally easy to read for people who are familiar with the related areas.

Major weaknesses:
- The performance gain is insignificant.
- Removal of 1D-convolution may hurt if the dataset has ordered channels. Whether or not a dataset has ordered channels may not be apparent in some real-world applications.
- It is not clear whether the proposed techniques also work for other Mamba variants.

The two proposed techniques are simple, but we do not think it is right to reject a paper simply because of the simplicity of the proposed method. A simple idea could lead to paper acceptance if its effectiveness is thoroughly justified beyond a reasonable doubt. However, from the experiments, we are not convinced that the proposed method can lead to significant performance gain. A more comprehensive study by incorporating the suggestions of some reviewers would make this paper more ready for publication.

**Additional Comments On Reviewer Discussion:**

The authors responded to the comments of the reviewers and engaged in discussions with them. Additional experiments were also conducted. Nevertheless, all reviewers still feel that this is a borderline paper. The only (minor) difference is which side of the borderline. It is not suitable for ICLR to accept a paper still with doubts that need to be sorted out. Addressing the issues, particularly the weaknesses listed above, will make this paper more ready for publication.

---

### Decision · Program_Chairs · 2025-01-22

Reject